# The role of Nrf2 in thyroid maturation and hormone synthesis in vertebrate models

Pierre Gillotay[1], Sushant Bangru[2], Benjamin Dassy[1], Benoit Haerlingen[1], Meghna P Shankar[1], Barbara F Fonseca[1], Panos G Ziros[3], Sumeet P Singh[1], Gerasimos P Sykiotis[3], Mirian Romitti[1,*], Sabine Costagliola[1,*]

In vertebrates, the thyroid gland synthesizes hormones that act on almost all tissues and are essential for normal growth and metabolism. Thyroid hormone production relies on iodination of thyroglobulin and requires $H_2O_2$, which contributes to a relatively high basal oxidative stress in the thyroid that must be tightly controlled to prevent cellular damage. The thyroid has efficient antioxidant and detoxifying enzymes that help it resist $H_2O_2$-induced oxidative stress maintaining the homeostasis necessary for hormone synthesis. By regulating the expression of genes involved in cellular detoxification, NRF2 acts as a master regulator of the cellular defense against oxidative stress. Using zebrafish embryos and mouse ESC-derived thyroid organoids, we generated nrf2a/Nrf2 loss-of-function and identified a common dyshormonogenesis phenotype. Although in zebrafish, the driving mechanisms are possibly related to thyroglobulin iodination defects, in thyroid organoids, it is likely due to a reduction in Tg production, consequently affecting folliculogenesis and thyroid hormone production.

## Introduction

Protection against oxidative stress (O.S.)–induced damage is a common feature of every living organisms (recently reviewed by Siauciunaite et al [2019]) and is mediated by a wide range of biological processes including (but not limited to) glutathione-induced gene expression (Hayes & McLellan, 1999), and regulation of NADPH levels via the pentose phosphate pathways (Ralser et al, 2007) and the superoxide dismutase (Muid et al, 2014) and catalase (Chelikani et al, 2004) activity. In addition, in many species, the nuclear factor erythroid 2-like 2 transcription factor (NRF2, encoded by the NFE2L2 gene in humans) acts as a master regulator of the cell's oxidative and metabolic stress defense system (Mukaigasa et al, 2012; Loboda et al, 2016; Thanas et al, 2020). In physiological conditions, Nrf2 is sequestered in the cytoplasm by its repressor the Kelch-like ECH-associated protein 1 (Keap1). To efficiently repress Nrf2, Keap1 forms a complex with Cul3 to recruit an E3 ubiquitin ligase complex that will drive the poly-ubiquitination and subsequent proteasomal degradation of Nrf2. Under stress conditions, the conformation of the Keap1-Cul3 complex will change, preventing ubiquitination and degradation of the newly formed Nrf2 protein, therefore allowing its translocation to the nucleus to regulate detoxifying and antioxidant gene expression (Iso et al, 2016; Yamamoto et al, 2018; Renaud et al, 2019). To regulate gene expression, Nrf2 will bind to the antioxidant response element (ARE) located in the promoter or enhancer region of target genes (Rushmore et al, 1991; Zhu et al, 2016).

Considering its functional need for $H_2O_2$, the regulation of O.S. in the thyroid tissue has attracted growing attention over the last few years (Renaud et al, 2019; Thanas et al, 2020). In all vertebrates, the thyroid gland plays a primordial role during development and acts as a central regulator of the physiology of any individual (Yen et al, 2006; Mullur et al, 2014; Ortiga-Carvalho et al, 2016). During embryonic development, the thyroid, through the production of thyroid hormones, facilitates the production of growth hormones (Shields et al, 2011; Liu et al, 2015) and plays a critical role in brain organogenesis and maturation (Mohan et al, 2012; Moog et al, 2017). In adults, it controls basal metabolism, cardiac function, and body temperature among other functions (Klein & Ojamaa, 2001; Kahaly & Dillmann, 2005; Kim, 2008; Mullur et al, 2014; Ortiga-Carvalho et al, 2016). Congenital hypothyroidism (CH) is the most common congenital endocrine disorder (affecting 1 in every 3,500 births) and is characterized by a lack of physiological thyroid function at birth. This pathology can be caused by a defect affecting either the thyroid organogenesis or the thyroid hormone synthesis per se (Wassner, 2018). If left untreated, CH will cause severe mental and growth retardation among other pathological consequences (Persani et al, 2018). Thyroid functions are mediated through both triiodothyronine (T3—biologically active but produced to a lesser extent) and thyroxine (T4—biologically less active but produced in high quantity), which are the main hormones produced (Carvalho

[1]Institut de Recherche Interdisciplinaire en Biologie Humaine et Moléculaire (IRIBHM), Université Libre de Bruxelles (ULB), Brussels, Belgium   [2]Duke Cell Biology department, Duke University, Durham, NC, USA   [3]Service of Endocrinology, Diabetology and Metabolism, Lausanne University Hospital and University of Lausanne (UNIL), Lausanne, Switzerland

Correspondence: mirian.romitti@ulb.be; sabine.costagliola@ulb.be
*Mirian Romitti and Sabine Costagliola contributed equally to this work

& Dupuy, 2017). Thyroid hormone (TH) production is an evolutionary conserved multistep process ultimately leading to the $H_2O_2$-dependent coupling of iodine and thyroglobulin, thereby creating iodinated thyroglobulin (Tg-I), the precursor of the thyroid hormones (Di Jeso & Arvan, 2016; Carvalho & Dupuy, 2017). Consequently, as soon as the TH production machinery is functional, the thyroid gland undergoes a higher basal level of $H_2O_2$-induced oxidative stress than other tissues.

Although a minimal level of O.S. is mandatory for thyroid cells' growth, function, and proliferation (Poncin et al, 2009, 2010), if left unchecked, a higher level of $H_2O_2$ will break the balance between oxidant product and thyroid's antioxidant defenses, ultimately resulting in O.S.-induced damage that can lead to genomic and epigenetic mutation (El Hassani et al, 2019; Renaud et al, 2019).

Recent studies performed on adult mouse thyroid glands demonstrated the role of Nrf2 as a direct controller of thyroglobulin (Tg) expression and as a central actor of the thyroid gland stress defense system (Renaud et al, 2019; Chartoumpekis et al, 2020). Despite these important studies evaluating the oxidative stress in functional thyroid follicular cells (TFCs), we lack information on the potential role of *Nrf2* during thyroid gland development and its maturation process. Here, using zebrafish embryos and mouse embryonic stem cell (mESC)–derived thyroid follicles, we sought to characterize the role of *Nrf2/nrf2a* during mammalian and nonmammalian thyroid development and maturation.

# Results

## Identification of *nrf2a* as an actor of zebrafish thyroid functional maturation

Because of whole genome duplication, the zebrafish genome encodes for two orthologs of mammalian Nrf2: *nrf2a* (encoded by the *nfe2l2a* gene, ZDB-GENE-030723-2) and *nrf2b* (encoded by the *nfe2l2b* gene, ZDB-GENE-120320-3), both encoding for transcription factors (TFs) (Timme-Laragy et al, 2011). To assess whether one of these two paralogs plays a role during thyroid development or thyroid functional maturation, we generated single-guide RNA (sgRNA) targeting the DNA binding domain of *nrf2a* and *nrf2b*, located on their exon 5 and 4, respectively. To uncover potential thyroid developmental defects, we took advantage of our previously published F0 CRISPR/Cas9-based screening approach (Trubiroha et al, 2018). Here, after one-cell-stage injection in Tg(*tg:nlsEGFP*) embryos (Trubiroha et al, 2018), thyroid development was monitored in vivo at 55, 72, and 144 hours post-fertilization (hpf) for putative developmental defect affecting either early thyroid development, proliferation, migration, or thyroid functional maturation, respectively. In these experiments, two controls were used: (1) the injection of a sgRNA targeting *adamtsl2* as a control for false-positive phenotypes (Trubiroha et al, 2018) and (2) the analysis and raising of noninjected embryos as a fertility and development quality control. At 55 and 80 hpf, the thyroid development of the nrf2a- and nrf2b-injected embryos was similar and no gross developmental defects were observed compared with their *adamtsl2* crispants and noninjected siblings (Fig S1A–H).

However, at 144 hpf, thyroid tissue appeared enlarged in around 55% (170/312) of the *nrf2a* crispants compared with their *adamtsl2* crispants and noninjected siblings (Fig S1I–L). Conversely, we could not uncover any gross or thyroid-related developmental defect in *nrf2b* F0 crispants compared with their *adamtsl2* crispants and noninjected siblings despite the high mutagenic activity of the sgRNA used in these experiments. Considering the reproducibility of the observed phenotype in F0, we decided to generate a stable nrf2a mutant line to further investigate the mechanism underlying the observed developmental defect of the thyroid gland.

### Generation of a stable *nrf2a* mutant line

To assess the transmission of the mutated alleles and associated thyroid phenotype, we raised F0 Tg(*tg:nlsEGFP*) *nrf2a* crispants to adulthood and outcrossed them with AB* WT animals to identify and select transmitted mutated alleles. F1 progeny was raised to adulthood and subsequently genotyped to characterize the transmitted mutated alleles using Sanger sequencing. We identified several mutated *nrf2a* alleles harboring insertion, deletion, or a mix of both and selected an allele presenting a deletion of 5 nucleotides between the nucleotides 1450 and 1454 of the reference coding sequence, this deletion is in the exon 5 before the DNA binding domain of the nrf2a protein (hereafter referenced as *nrf2aΔ5*) (Fig 1A). The *nrf2aΔ5* allele is predicted to induce a frameshift leading to a truncated Nrf2a protein of 550 amino acids (aa), whereas the WT protein is 587 amino acids long, including 67 incorrect amino acids at the C terminus. In silico analysis of the mutated protein encoded by the *nrf2aΔ5* allele revealed that out of the 112 amino acids composing the WT protein coiled-coil domain in which the DNA binding region resides, only eight amino acids were conserved in the mutated protein, suggesting that it is nonfunctional. To experimentally assess the functional consequences of our *nrf2aΔ5* mutated allele, we tested its ability to activate the transcription of an ARE-driven firefly luciferase reporter gene following a previously published protocol (Mukaigasa et al, 2012). For this, the comparison was made with the nrf2a WT protein and the previously published nrf2a fh318 mutated protein known to be nonfunctional (Mukaigasa et al, 2012). After transient transfection in HEK cells, WT nrf2a strongly activates the transcription of the firefly luciferase reporter gene, whereas both the *nrf2aΔ5* and *nrf2a* fh318 mutants induced much weaker expression of the reporter gene (Fig 1B). This validates that the *nrf2aΔ5* allele, like the *nrf2a* fh318 mutant allele, encodes a nonfunctional TF.

### *nrf2a* loss-of-function induces hypothyroidism in homozygous mutant zebrafish embryos

We then analyzed, in vivo, the thyroid phenotype of Tg(*tg:EGFPnls*) *nrf2aΔ5* homozygous embryos to see whether they exhibited a thyroid phenotype similar to those observed in F0 crispants. As for F0 crispants, no thyroid developmental defect was observed at 55 (Fig 1D and G) and 80 hpf (Fig 1E and H) in *nrf2aΔ5* hemi- and homozygous embryos and their WT siblings. However, at 144 hpf, the thyroid gland of *nrf2aΔ5* homozygous embryos appears enlarged along the anteroposterior axis compared with the gland of their hemizygous mutant and WT siblings (Fig 1F and I). To quantify

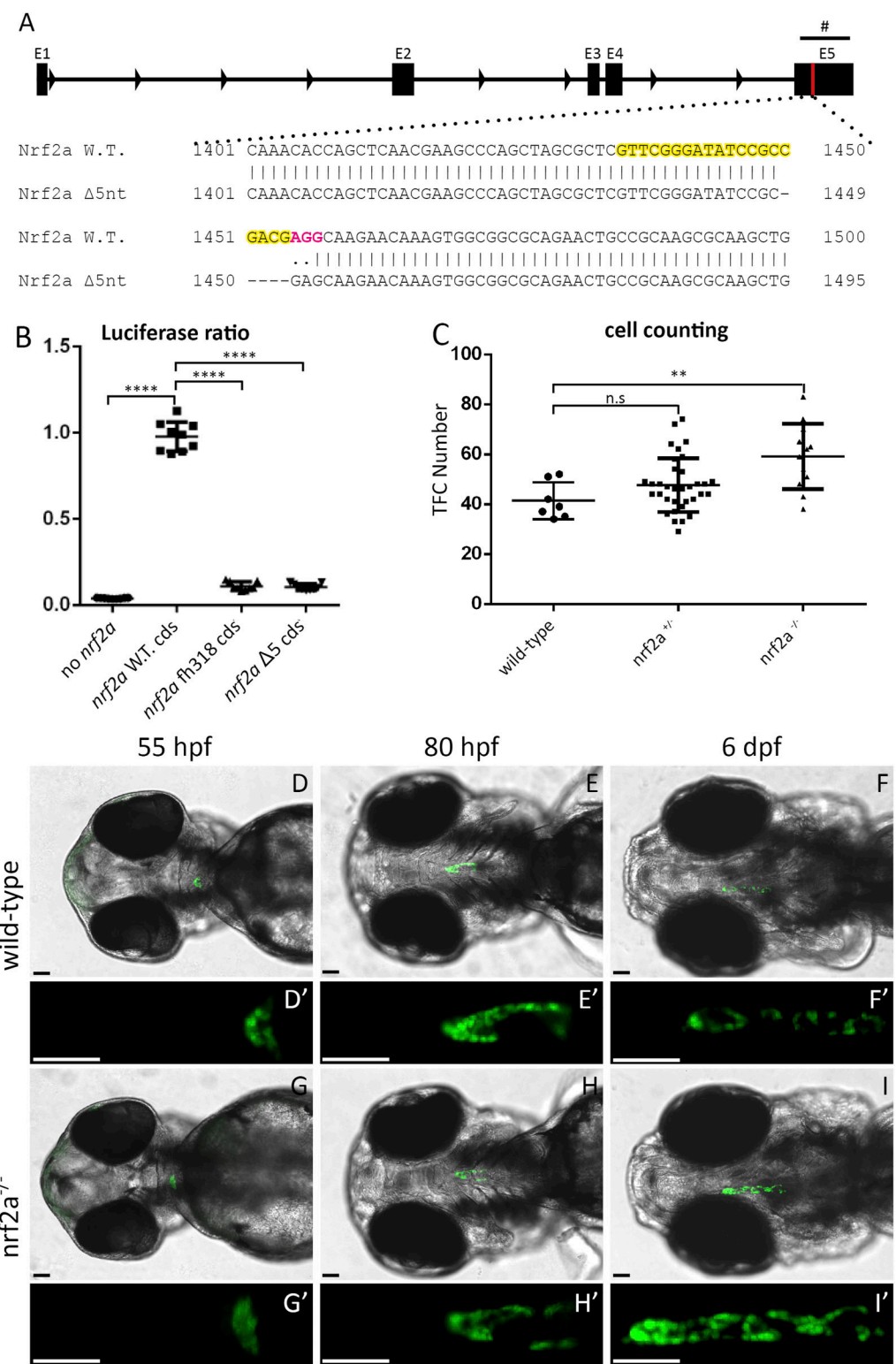

**Figure 1. Live imaging of transgenic Tg(tg:nlsEGFP) zebrafish embryos carrying a 5-nt deletion in *nrf2a* allows real-time in vivo analysis of thyroid development.**
**(A)** 5-nt deletion was generated in the exon 5 (E5) of the nrf2a gene resulting in a frameshift-inducing premature stop codon disturbing the DNA binding domain of the gene. The guide sequence is highlighted in yellow with the PAM sequence highlighted in magenta. **(B)** Activation of an ARE-driven luciferase reporter by WT and mutated nrf2a. HEK cells were transfected with plasmids encoding for the full-length coding sequences of a WT nrf2a, nrf2a fh318, or nrf2a Δ5 to evaluate their respective ability to drive luciferase expression. Results show that WT nrf2a (second from the left) can drive luciferase expression, whereas nrf2a fh318 and Δ5 cannot drive luciferase expression (third and fourth from the left, respectively). Results are shown as the mean ± SD. The asterisk denotes significant differences between positive control (WT nrf2a cds) and test conditions (****$P$ < 0.0001, Dunnett's multiple comparison test). **(C)** Quantification of thyroid follicular cell (TFC) number in individual WT (n = 7), heterozygous (n = 34), and homozygous mutant embryos (n = 12) shows a significant difference between WT and homozygous mutant. Results are shown as the mean ± SD.

this apparent thyroid enlargement, we used confocal live imaging on progeny derived from an in-cross of Tg(tg:nlsEGFP); nrf2aΔ5 hemizygous adult animals to perform thyroid cell counting at 144 hpf (Fig 1C). On average, thyroid glands of WT embryos were composed of 41.4 ± 7.3 cells, whereas thyroids of homozygous nrf2aΔ5 embryos were composed of 59.2 ± 13.1 cells, confirming a 40% enlargement of their thyroid gland. Excessive proliferation of the TFCs is often associated with continuous stimulation of the thyroid gland by the TSH signaling pathway. Such continuous stimulation may result from a defect in thyroid hormone (TH) production, triggering the feedback loop controlling the hypothalamus–pituitary–thyroid axis (Dumont et al, 1992; Ortiga-Carvalho et al, 2016). To test this hypothesis, we performed whole-mount immunofluorescence (WIF) for thyroxine (T4) and its precursor the iodinated thyroglobulin (Tg-I) on 6 dpf progeny from nrf2aΔ5 heterozygous carrier (Fig 2A–O). WIF staining for T4 and Tg-I confirmed that the thyroid enlargement observed in nrf2aΔ5 homozygous embryos coincides with dyshormonogenesis as shown by the lack of or barely detectable T4 and Tg-I staining (6.75 × $10^5$ ± 2.68 × $10^5$ A.U for T4 and 1.36 × $10^6$ ± 6.61 × $10^5$ A.U for TG-I) compared with their WT siblings (2.10 × $10^6$ ± 6.13 × $10^5$ A.U for T4 and 3.37 × $10^6$ ± 1.50 × $10^6$ A.U for TG-I) (Fig 2Q and R). To ensure that the lack of T4 was not caused by a defect in thyroid follicle cell polarization, we performed immunostaining of ZO-1, a protein that labels the apical junctions between the TFC. Confocal imaging of WT and homozygous nrf2aΔ5 embryos revealed no differences in follicular polarization between embryos (Fig S2A–F), suggesting that the observed deficiency of T4 production is not caused by a defect in thyroid follicle morphogenesis per se.

### Loss of nrf2a function does not impair the expression of thyroid maturation genes

The phenotype of thyroid dyshormonogenesis observed in homozygous nrf2aΔ5 mutants may be due to several causes. Indeed, T.H. synthesis requires complex machinery allowing uptake and transport of iodide from the bloodstream to the colloidal space, coupling of iodine to the thyroglobulin, and eventually proteolysis of iodinated thyroglobulin and release of T.H. in the bloodstream (Carvalho & Dupuy, 2017). As immunostaining for T4 and Tg-I reveals an absence of both signals in nrf2aΔ5 homozygous mutants, we decided to analyze the expression of thyroid functional markers and stimulating factors on 6 dpf nrf2aΔ5 zebrafish embryos (Fig 2S). We first looked at the thyroid-stimulating signaling by analyzing the expression of tsh-β and tshr. In nrf2aΔ5 homozygous mutants, the expression of tsh-β was greatly increased compared with their WT siblings (Fig 2S, second from the left), consistent with the known effects of a lack of T.H. production (Shupnik et al, 1985; Chiamolera & Wondisford, 2009). On the other hand, no differences in the tshr expression were found in nrf2aΔ5 homozygous mutant embryos

compared with their WT siblings (Fig 2S). As the absence of Tg-I staining in nrf2aΔ5 homozygous mutant embryos suggests a defect in the first steps of T.H. synthesis, we evaluated the slc5a5, tpo, duox, and tg expression levels, critical components in the formation of Tg-I (Carvalho & Dupuy, 2017). Consistent with the increase in tshβ expression (Opitz et al, 2011), slc5a5 and tpo levels were also increased in nrf2aΔ5 homozygous mutants compared with their WT siblings (Fig 2S). However, duox levels were not significantly different from the controls, whereas tg expression, although not statistically significant, appeared slightly reduced in nrf2aΔ5 homozygous mutants compared with their WT siblings despite the increase in the TFC number (Fig 2S). To assess whether nrf2a could regulate tg at the posttranscriptional level, we performed tg whole-mount immunofluorescence in 6 dpf zebrafish embryos (Fig S3A–L). Interestingly, no changes in tg protein levels were observed among control (Fig S3A) and nrf2aΔ5 mutants (Fig S3B and C), suggesting that the lack of nrf2a does not significantly affect tg protein production and its posttranslational regulation. To assess whether the changes in mRNA levels were caused by defective nrf2a or by disrupted Tsh stimulation, we treated nrf2aΔ5 homozygous mutants and WT embryos with 1-phenyl-2-thiourea (PTU 30 μg/ml from 24 to 144 hpf), a chemical commonly used to inhibit pigmentation but also shown to inhibit T.H. production by affecting the thyroid–pituitary axis (Elsalini & Rohr, 2003; Opitz et al, 2011). Gene expression analysis showed that upon PTU treatment, tsh-β expression was increased in both WT and nrf2aΔ5 homozygous mutants, whereas slc5a5 expression increased in WT embryos compared with their nontreated siblings (Fig 2T and U), confirming the efficiency of PTU in inhibiting T.H. production by triggering the Tsh-mediated feedback loop. Although PTU treatment increased tg expression in WT embryos (Fig 2T), it remained unaffected in nrf2aΔ5 homozygous mutants compared with their nontreated siblings (Fig 2U). Interestingly, when comparing the modulation of tg expression in both PTU-treated WT and nrf2aΔ5 homozygous embryos, we observed a marked reduction in tg expression in nrf2aΔ5 homozygous mutant embryos (Fig 2V). These results suggest that the lack of functional nrf2a affects the modulation of tg expression in zebrafish thyroid only under stress conditions. Together, despite the conservation in the expression of the main maturation/functional genes in nrf2a KO embryos, we cannot rule out that the activity of key enzymes related to thyroglobulin iodination such as tpo and duox is not affected.

### The thyroid dyshormonogenesis in nrf2aΔ5 homozygous mutant embryos is a cell-autonomous phenotype

Although the initial characterization of the nrf2aΔ5 mutant phenotype indicates a role of nrf2a during thyroid functional maturation, we could not rule out the possibility that this phenotype is secondary to another defect affecting the whole zebrafish

---

The asterisk denotes significant differences between groups of embryos (**P < 0.01, Kruskal–Wallis multiple comparison test). **(D, E, F, G, H, I)** show pictures of the whole head of a representative embryo at a specified developmental stage (ventral view, anterior is to the left). **(D', E', F', G', H', I')** show an enlarged picture of the thyroid region of the corresponding embryo. Epifluorescence microscopy of live Tg(tg:nlsEGFP) zebrafish reveals an increase in the thyroid size at 6 days post-fertilization (dpf) in the mutant embryos compared with their WT and heterozygous siblings. Live imaging also reveals that before 4 dpf, thyroid development shows no apparent differences between mutant, heterozygous, and WT embryos. Scale bars: 50 μm (D, D', E, E', F, F', G, G', H, H', I, I'). "#" indicates the DNA binding domain of the nrf2a gene.

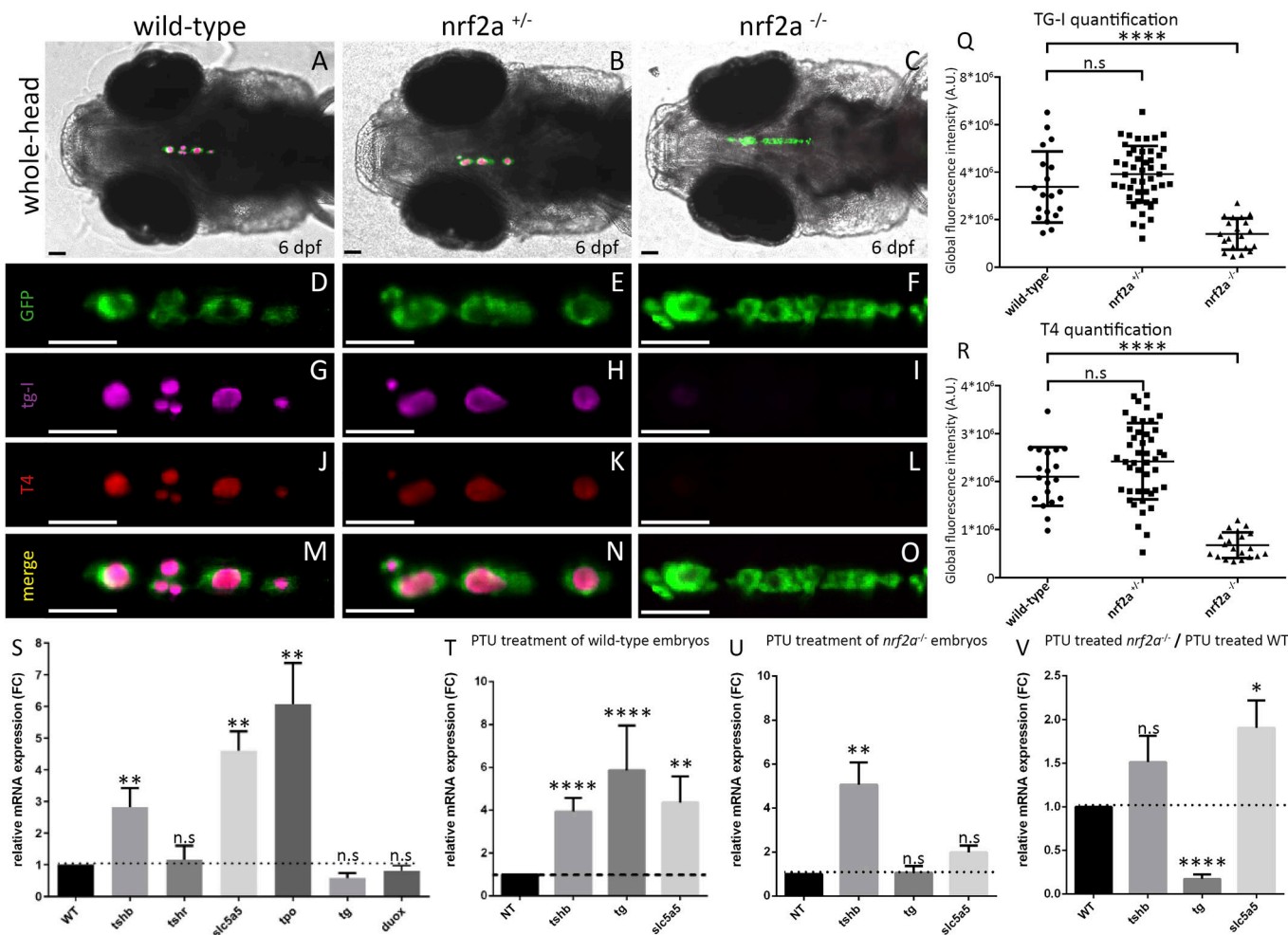

**Figure 2. Loss-of-function of *nrf2a* induces thyroid dyshormonogenesis.**
**(A, B, C, D, E, F, G, H, I, J, K, L, M, N, O)** Immunofluorescence on 6 dpf Tg(tg:nlsEGFP) zebrafish embryos targeting GFP, thyroxine (T4-Cy3, red), and iodinated thyroglobulin (TG-I–Cy5, magenta) reveals a defect in thyroid hormone production in *nrf2a* homozygous mutant compared with their WT and heterozygous mutant siblings. **(A, B, C)** show whole head pictures of representative embryos of each genotype (ventral view, anterior to the left). **(D, E, F, G, H, I, J, K, L, M, N, O)** Enlarged thyroid views of the corresponding embryos demonstrated thyroid follicular organization in all embryos (D, E, F); however, an absence of TG-I and T4 was observed only in *nrf2a⁻/⁻* embryos (F, I, L, O). **(Q, R)** Quantification of iodinated thyroglobulin (Q) and colloidal thyroxine (R) immunofluorescence signal in individual WT (n = 19), heterozygous (n = 45), and homozygous mutant embryos (n = 20) reveals a significant reduction in the *nrf2a⁻/⁻* group. Results are shown as the mean ± SD. The asterisk denotes significant differences between groups of embryos (****$P$ < 0.0001, one-way ANOVA multiple comparison test). Scale bars (A, B, C, D, E, F, G, H, I, J, K, L, M, N, O): 50 μm. **(S)** Quantitative analysis of thyroid marker expression levels by qRT-PCR in 6 dpf embryos revealed no differential expression of tshr ($P$ = 0.7706) and a slight, nonsignificant reduction of tg ($P$ = 0.1775) and duox ($P$ = 0.1) in nrf2a homozygous mutant embryos compared with WT siblings. In contrast, mRNA levels of *tshβ* ($P$ < 0.01), *slc5a5* ($P$ < 0.01), and *tpo* ($P$ < 0.01) were significantly increased in nrf2a mutant embryos, indicating elevated basal expression of thyroid function–related genes. **(T, U)** Quantitative analysis of thyroid marker expression in 6 dpf WT and nrf2a⁻/⁻ embryos after PTU treatment compared with their respective nontreated (NT) controls. In WT embryos, PTU treatment induced a strong increase in *tshβ* (****$P$ < 0.0001), *tg* (****$P$ < 0.0001), and *slc5a5* ($P$ < 0.01). In nrf2a⁻/⁻ embryos, PTU treatment significantly increased *tshβ* expression (**$P$ < 0.01), whereas *tg* ($P$ > 0.9999) and *slc5a5* ($P$ = 0.2499) expression did not significantly change relative to their NT controls. These panels represent within-genotype fold-change responses to PTU treatment, normalized to each genotype's own baseline expression level. **(V)** Direct comparison of thyroid marker expression between PTU-treated WT and PTU-treated nrf2a⁻/⁻ embryos. In this panel, expression levels are compared between genotypes under the same treatment condition, rather than relative to each genotype's baseline. Quantitative analysis showed that *tshβ* expression does not significantly differ between PTU-treated *nrf2a⁻/⁻* embryos and their WT siblings ($P$ > 0.9999). However, nrf2a⁻/⁻ embryos fail to up-regulate *tg* upon PTU treatment compared with WT (****$P$ < 0.0001), whereas *slc5a5* expression remains significantly higher in PTU-treated nrf2a⁻/⁻ embryos ($P$ = 0.0183). All qRT-PCR data represent relative gene expression normalized to the indicated control group (nontreated [NT] or WT, shown as the black column). The dashed black line indicates the control expression level to facilitate visual comparison across conditions. Data are presented as the mean ± SD. Statistical significance was determined using the Kruskal–Wallis test with multiple comparisons. The asterisk denotes significant differences between groups (*$P$ < 0.05, **$P$ < 0.01, ****$P$ < 0.0001). "n.s." denotes no significant difference. N varies between 5 and 6 independent pools of 10 embryos per gene.

embryos. To exclude non–cell-autonomous effects, we sought to rescue the thyroid function by generating a new transgenic line Tg(*tg:nrf2a^T2A_mKO2nls*) in which the thyroglobulin promoter drives the expression of a functional *nrf2a* coding sequence specifically in the TFCs. Furthermore, for visualization, the nrf2a coding sequence is linked, via a T2A viral peptide, to the nuclear-localized

monomeric Kusabira-Orange 2 fluorescent protein (hereafter referred to as the *nrf2a_mKO2* line or "rescue" allele). Although a fluorescent reporter allows easy selection of transgenic embryos, we wanted to confirm whether the overexpression of *nrf2a* mRNA could be detected in the thyroid of transgenic embryos by WISH. Indeed, because of its short half-life (McMahon et al, 2004), the detection of *nrf2a* mRNA in some tissues falls below the sensitivity of WISH staining. After WISH staining, *nrf2a* can be easily detected in tissues such as the developing gills and the liver but remains undetectable in the developing thyroid in 72 hpf WT embryos. In contrast, in addition to the other tissue, *nrf2a* can be detected in the thyroid of 72 hpf *nrf2a_mKO2* embryos, confirming that our transgenic line displays a thyroidal overexpression of *nrf2a* (Fig S4A and B).

Using this transgenic line, we generated animals carrying the rescue allele and the *nrf2aΔ5* mutation. The thyroid function was assessed among the rescued embryos at 6 dpf by T4 immunostaining as we had done previously (Fig 3A–P). In WT animals, quantification of the T4 fluorescent signal showed that the rescue transgene in the thyroid cells did not significantly affect T4 production ($2.62 \times 10^6 \pm 8.55 \times 10^5$ A.U. in WT embryos versus $2.40 \times 10^6 \pm 4.64 \times 10^5$ A.U. in WT *nrf2a_mKO2* embryos) (Fig 3Q). However, in 6 dpf *nrf2aΔ5* homozygous embryos, we could observe that T4 production was restored to physiological levels upon expression of the rescue allele ($2.34 \times 10^6 \pm 4.93 \times 10^5$ A.U. in rescued *nrf2aΔ5* homozygous embryos; $2.62 \times 10^6 \pm 8.55 \times 10^5$ A.U. in WT embryos; and $1.67 \times 10^6 \pm 5.72 \times 10^5$ A.U. in nonrescued *nrf2aΔ5* homozygous embryos) (Fig 3Q). In addition, in the presence of the rescue allele, the thyroid of *nrf2aΔ5* homozygous mutant embryos appeared to have a comparable size to WT *nrf2a_mKO2* embryos. Cell counting on 6 dpf *nrf2aΔ5* and WT embryos carrying or not the rescue allele revealed that the presence of the rescue allele in WT embryos does not affect the thyroid cell number (Fig 3R). However, although the thyroid glands of *nrf2aΔ5* homozygous embryos were, on average, composed of 60.4 ± 13.9 cells, homozygous mutant embryos carrying the rescue allele had 43.4 ± 10 cells (Fig 3R). Altogether, these data support the cell-autonomous nature of thyroid dyshormonogenesis phenotype observed in *nrf2aΔ5* homozygous mutant zebrafish embryos.

Using zebrafish embryos, we showed that impairing the function of *nrf2a* strongly affects thyroid functional maturation, leading to a lack of T.H. production. We also proved that such a defect does not result from a disorganization of the thyroid follicular structure or a lack of expression of genes involved in the T.H. production machinery such as *tg*, *slc5a5*, *duox*, and *tpo*. In addition, we demonstrated that *nrf2a*-induced dyshormonogenesis is a cell-autonomous phenotype that does not result from a lack of the thyroglobulin protein.

## Loss of *Nrf2* impairs differentiation of mESC-derived thyroid follicles

Recently, the role of *Nrf2* in adult thyroid gland physiology has been investigated using a ubiquitous and thyroid-specific *Nrf2* Knockout mouse model (Ziros et al, 2018). In this study, the authors demonstrate that adult mice lacking *Nrf2* are euthyroid, which is strikingly different from our results in zebrafish embryos. Mouse

pups develop in utero, and as such, follicular development can be affected not only by the surrounding tissues (e.g., endothelium) but also by maternal factors. To examine whether the absence of Nrf2 can impact TFC differentiation and/or folliculogenesis in a manner independent of such factors, we used the in vitro model of mESC-derived thyroid follicles that we previously developed (Antonica et al, 2012; Romitti et al, 2021). mESC-derived thyroid follicles offer the unique advantage of recapitulating the main aspects of thyroid development in an ex vivo controlled medium.

The mouse ortholog of the zebrafish *nrf2a* is encoded by the *Nfe2l2* gene (also known as *Nrf2*) and is composed of five exons. Similar to the zebrafish *nrf2a*, the DNA binding domain is located within the fifth exon. Thus, to generate a loss-of-function *Nrf2* equivalent to the zebrafish mutant line, we targeted a region immediately upstream of the DNA binding domain using the CRISPR/Cas9 technology. After targeted mutagenesis, we identified two clones carrying mutations that lead to disruption of the DNA binding domain and a premature STOP codon. The first clone (hereafter referred to as "clone 16" or *Nrf2* KO; see Fig 4A) harbors a deletion of 19 nucleotides, whereas the second clone (hereafter referred to as "clone 19") harbors a deletion of one nucleotide.

We firstly validated the ability of both clones to overexpress Nkx2.1 and Pax8 after Dox treatment on the seventh day of the differentiation protocol using immunostaining, showing that the *Nrf2* loss-of-function does not affect the tetracycline-inducible overexpression of *Nkx2.1* and *Pax8*.

Both clones were then differentiated into thyroid organoids following our previously published protocol (Antonica et al, 2012; Romitti et al, 2021) to assess their ability to differentiate, form follicles, and produce T.H. (day 22). Immunofluorescence was performed at day 22 to detect the expression of thyroid markers (Nkx2.1 and Tg). Although both *Nrf2* KO clones can express Nkx2.1, they showed strong impairment in Tg expression (Fig S5A–C) and variable capability to organize into follicles. Considering that both clones presented similar phenotypes, we focused our investigations on clone 16; therefore, all following data and mention of the *Nrf2* KO refer to a clone 16 line. Analysis of this mutation revealed that it leads to the appearance of a premature stop codon resulting in a truncated protein of 469aa length, of which only the first 441aa are conserved with the 597–aa-long WT protein.

To better characterize the *Nrf2* KO–driven phenotype, we analyzed the expression of thyroid markers on day 7 of our differentiation protocol. Compared with the control line, *Nrf2* KO cells expressed similar levels of *Pax8*, *Tshr*, and exogenous and endogenous *Nkx2.1* (Fig 4B). However, *Tg* expression was significantly down-regulated (Fig 4B). At the end of the protocol (day 22), Nrf2 KO cells displayed higher expression levels of *Nkx2.1*, *Pax8*, *Tshr*, *Nis*, *Tpo*, *Duox2*, and *Duoxa2*, whereas *Tg* mRNA was consistently lower (Fig 4C). Remarkably, we also observed a decrease in the expression of *Gstp1* and *Gclc* (Fig 4C), two known targets of Nrf2 (Mukaigasa et al, 2012; Renaud et al, 2019) validating that the 19-bp deletion abrogates Nrf2 function. Flow cytometry analysis demonstrated a higher percentage of Nkx2-1+ cells at the Nrf2 KO condition (65.3%) than the WT control (27.1%). Moreover, we observed that the proportion of bTg-eGFP+ cells among Nrf2 KO NKX2-1+ cells was significantly lower (6.1%) than in the control (37.4%; Fig 4D).

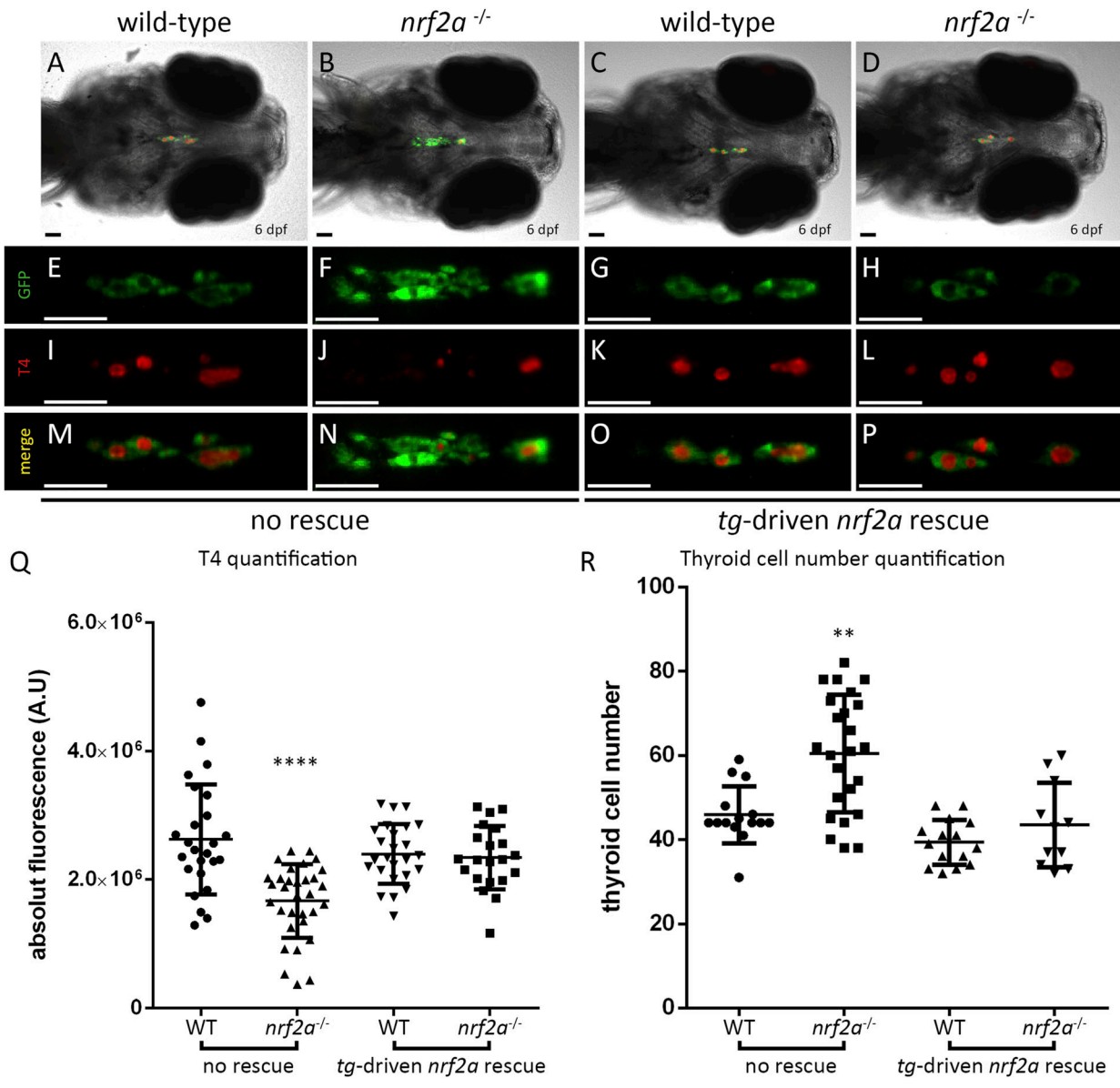

**Figure 3. Thyroid-specific rescue of nrf2a function is sufficient to recover physiological T4 production.**
**(A, B, C, D, E, F, G, H, I, J, K, L, M, N, O, P)** Immunofluorescence on 6 dpf Tg(*tg:nlsEGFP*; *tg:nrf2a*$^{T2A\_mKO2}$) zebrafish embryos targeting GFP and thyroxine (T4-Cy3, red) reveals that the presence of the *tg:nrf2a*$^{T2A\_mKO2}$ construct is sufficient to rescue the thyroid hormone production defect in *nrf2a* homozygous mutants compared with their WT and heterozygous mutant siblings (A, B, C, D, E, F, G, H, I, J, K, L, M, N, O, P). **(A, B, C, D)** show whole head pictures of representative embryos of each genotype (ventral view, anterior to the left). **(E, F, G, H, I, J, K, L, M, N, O, P)** Enlarged views of the thyroid of the corresponding embryos demonstrate thyroid follicular organization in all embryos (E, F, G, H) and reveal that the absence of T4 staining observed in nrf2a homozygous mutant embryos (second column to the left) is rescued in nrf2a homozygous mutant embryos expressing de *tg:nrf2a*$^{T2A\_mKO2}$ transgene (last column to the left). Scale bars: 50 $\mu$m. **(Q)** Quantification of colloidal thyroxine immunofluorescence signal in individual WT (n = 25), *nrf2a* homozygous mutant embryos (n = 32), WT *nrf2a*$^{T2A\_mKO2}$ (n = 25), and *nrf2a* homozygous mutant *nrf2a*$^{T2A\_mKO2}$ embryos (n = 21) confirmed the rescue of the T4 production in the *nrf2a* homozygous mutant *nrf2a*$^{T2A\_mKO2}$ embryo group compared with their siblings from other genotypes. Results are shown as the mean ± SD. The asterisk denotes significant differences between groups of embryos (****$P < 0.0001$, one-way ANOVA multiple comparison test). **(R)** Quantification of thyroid follicular cell (TFC) number in individual WT (n = 15) and homozygous *nrf2a* mutant embryos (n = 24) and WT (n = 15) and homozygous *nrf2a* mutant (n = 11) *nrf2a*$^{T2A\_mKO2}$ embryos shows that the *nrf2a*$^{T2A\_mKO2}$ transgene rescued the thyroid cell number in homozygous *nrf2a* mutant embryos. Results are shown as the mean ± SD. The asterisk denotes significant differences between groups of embryos (**$P < 0.01$, Kruskal–Wallis multiple comparison test).

Despite the increased proportion of Nkx2.1⁺ cells (Fig 4D), closer examination of follicular organization revealed qualitative differences in follicular architecture in Nrf2 KO cultures. Compared with controls, Nrf2 KO–derived cells exhibit both disorganized and follicular arrangements; however, the follicles that do form are more heterogeneous in size and shape and are often smaller. These observations indicate increased morphological variability in follicle organization under microscopic examination; however, because of technical limitations inherent to 3D cultures, precise quantification is not feasible, precluding definitive conclusions

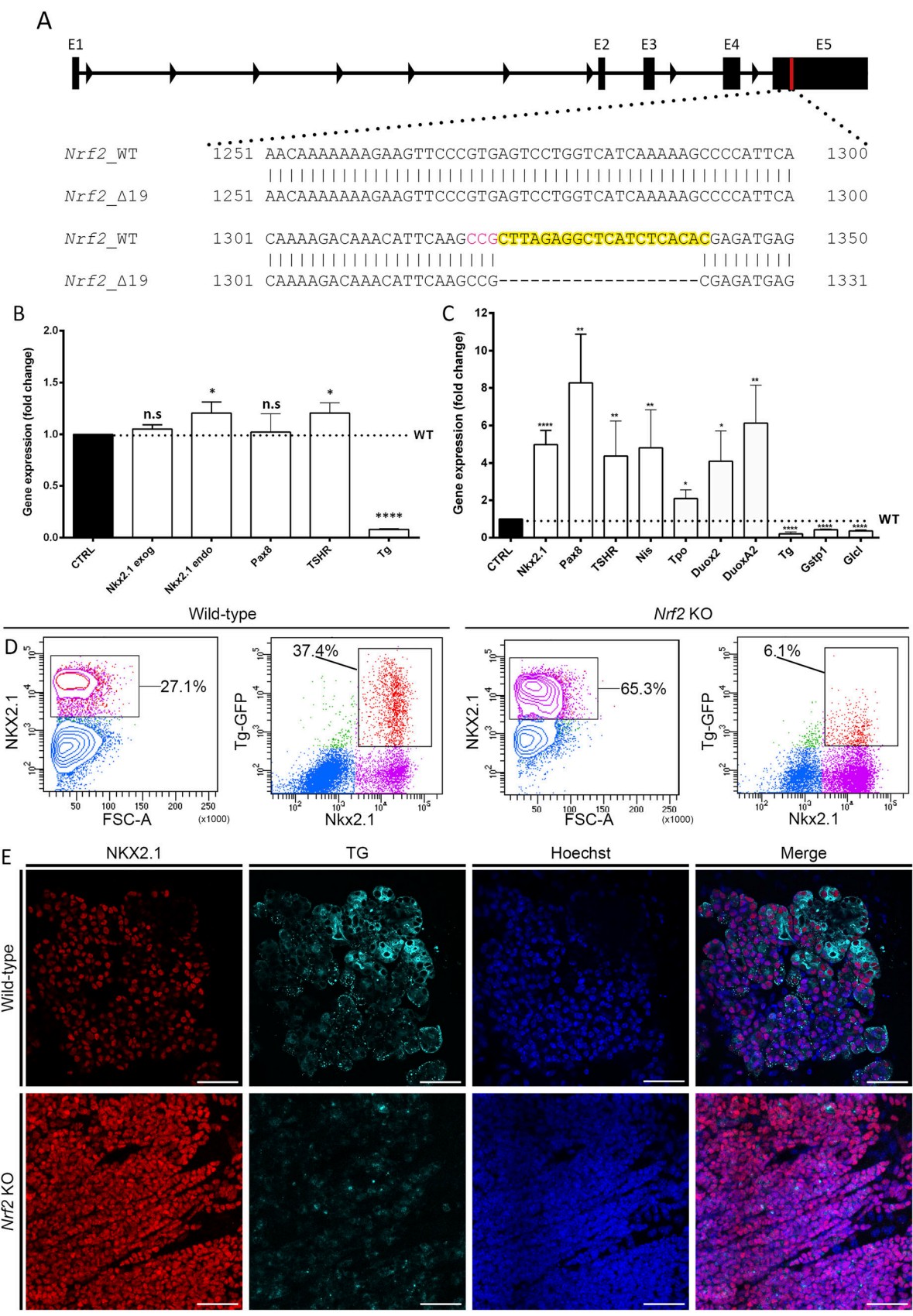

regarding the efficiency or overall integrity of folliculogenesis (Figs 4E and 5B and C). Immunostaining experiments also highlight the decreased Tg protein levels in Nrf2 KO–differentiated cells (Figs 4E and S5), while showing a greater proportion of Nkx2-1 cells compared with the control cell line (Fig 4D and E). The lower levels of Tg secretion could explain the impairment in follicles' proper formation in Nrf2 KO thyroid organoids.

### Functional Nrf2 is essential for thyroid hormone synthesis in mESC-derived follicles

The most important phenotype driven by the lack of Nrf2 is the absence of the Tg-I and T.H. production. To better explore this phenotype in our organoids, we analyzed the presence of Nis at the basal membrane of follicles and the ability of Nrf2 KO cells to produce iodinated thyroglobulin. We observed that the Nis protein was well expressed at the basolateral membrane of both Nrf2 KO and control cell lines at day 22. However, the presence of Tg-I was drastically reduced in *Nrf2* KO–derived follicles compared with WT controls (Fig 5A). This impairment likely reflects both decreased Tg expression and the altered, more variable follicular organization observed under Nrf2 KO conditions (Figs 4E and 5B and C). Iodine organification assays (Antonica et al, 2017) confirmed the observations made by Tg-I immunostaining. Indeed, although both cell lines could similarly uptake iodine (Fig 5D), the levels of protein-bound $^{125}$I were significantly lower in *Nrf2* KO thyroid cells (Fig 5D), resulting in a lower proportion of *Nrf2* KO–derived thyrocytes (3.43%) able to organify iodide compared with the control condition (22.45%) (Fig 5D–F). Conversely to the *nrf2aΔ5* zebrafish mutants, lack of Tg in mESC-derived thyroid organoids appears to be the primary cause of this defect in thyroid hormone production.

### Loss of *Nrf2* profoundly affects the transcriptional and chromatin accessibility landscape of mESC-derived thyrocytes

To understand the molecular effect of the loss of *Nrf2* on mESC-derived thyrocytes, we combined RNA and ATAC sequencing to uncover the mechanism underlying the phenotype. Importantly, although our mESC lines (*Nrf2* WT and KO) expressed the bTg-GFP reporter construct, down-regulation of *Tg* expression in Nrf2 KO–derived cells prevented us from sorting the thyroid cells. Therefore, RNA-sequencing experiments were performed with a mixed cell population. Transcriptomic analysis revealed significant differences between expression signatures of *Nrf2* WT and KO cells,

showing 1,566 up-regulated and 5,272 down-regulated genes in the *Nrf2* KO condition compared with the WT (Fig 6A). Among the differentially expressed genes, transcripts related to thyroid function were up-regulated (Fig S6A) whereas *Nrf2* target genes and *Nrf2* regulating genes were shown down-regulated (Fig S6A and D) (Raghunath et al, 2018). Gene ontology (GO) classification of the most differentially expressed genes revealed that *Nrf2* KO cells express mesodermal markers, as well as inflammation and stress-related gene signature (TNF-α, NF-κB, IL-2, and IL-6; Fig S6B and C), consistent with previously described results in adult *Nrf2* KO mice (Chartoumpekis et al, 2020). Aiming to identify other key genes potentially involved in the impairment of thyroid function, differential expression analysis revealed a set of TFs differentially expressed in *Nrf2* KO cells (Fig 6B). Down-regulation of Nrf2 target genes, exhibiting an antioxidant responsive element (ARE) within their promoter and/or regulatory regions, was also observed in the *Nrf2* KO condition (Fig S6D). Interestingly, identified TFs show strong interaction (Fig S6E) and are involved in oxidative stress response, AP-1 network, and activation of several signaling pathways such as MAPK, TNF-α, NF-κB, and TGF-β. To better understand the transcriptional mechanisms, we performed ATAC sequencing to assess chromatin landscape changes occurring upon *Nrf2* loss-of-function. Genome-wide analysis of ATACseq libraries from WT and NRF2 KO cells revealed a relatively higher incidence of open chromatin at the TSS of coding genes upon loss of Nrf2 (Fig 6D). Furthermore, although the large proportion of open chromatin peaks was common between the two conditions, a significant proportion of peaks were enriched in the *Nrf2* KO condition (Fig 6E). We also observed a positive correlation between RNA expression and chromatin accessibility of loci for many genes expressed in the thyroid gland as exemplified by more closed chromatin at *Tg* loci and open chromatin at *Pax8* loci (Fig 6F). The reduction in chromatin accessibility at Tg loci reinforces the role of Nrf2-dependent mechanisms in regulating its expression levels in mESC-derived organoids. Upon unbiased motif enrichment analysis, we identified enrichment for FOS: JUNB and STAT family TF motifs (Fig 6G) in *Nrf2* KO cells. Interestingly, these factors (*FosB*, *Stat1*, *Stat5a*) were not only up-regulated at the RNA level (Fig 6G), but they also displayed the presence of the ARE sequence in their promoter (at −355, −514, and −780 bp from the transcription start site of *Stat5a*; and at −120, −450, and −930 bp from the transcription start site of *Fosb*). Lastly, the up-regulation of *Stat* TFs in the *Nrf2* KO cells might play a role in the observed phenotype because more than 50% of genes up-regulated in ($\log_2$FC > 2) in this condition also

---

**Figure 4. Studies of Nrf2 KO mESC-derived thyroid organoids highlight the importance of Nrf2 during mammalian thyroid development.**
**(A)** 19-nt deletion was generated in the exon 5 (E5) of the Nrf2 gene resulting in a frameshift-inducing premature stop codon disturbing the DNA binding domain of the gene. The guide sequence is highlighted in yellow with the PAM sequence highlighted in magenta. **(B)** qRT-PCR analysis performed on day 7 of the differentiation protocol demonstrates significant down-regulation in *Tg* gene expression and slight up-regulation in *Nkx2.1* and *Tshr* gene expression. **(C)** qRT-PCR analysis performed at the end of the differentiation protocol (at day 22) demonstrates significant up-regulation in *Nkx2.1*, *Pax8*, *Tshr*, *Nis*, *Tpo*, *Duox2*, and *Duoxa2* gene expression and significant down-regulation in *Tg*, *Gstp1*, and *Gclc* gene expression. All qRT-PCR data show relative gene expression compared with their respective control (NT, nontreated; or −WT, wild type) represented by the black column. The dashed black line shows the control expression level across the panel to allowing visual comparison of the gene expression with their control. All statistical analyses were performed using the Mann–Whitney test. The asterisks denote a significant difference compared with the WT control: *P < 0.05, **P < 0.01, and ****P < 0.001. **(D)** Flow cytometry analysis of WT (first and second panels from the left) and Nrf2 KO (third and fourth panels from the left) thyroid organoids after 22 d of differentiation. First and third panels show the proportion of NKX2.1+ cells in the whole sample, and the percentages (%) of NKX2.1+ cells are displayed. Second and fourth panels show the proportion of bTg+ cells within the Nkx2.1+ cell population, and the percentages (%) are displayed. **(E)** Confocal images obtained after immunofluorescence experiments on fixed samples after 22 d of differentiation. Samples were marked for Nkx2.1 (red) and Tg (cyan). Nuclei were labeled using Hoechst (Blue). Scale bars: 100 μm.

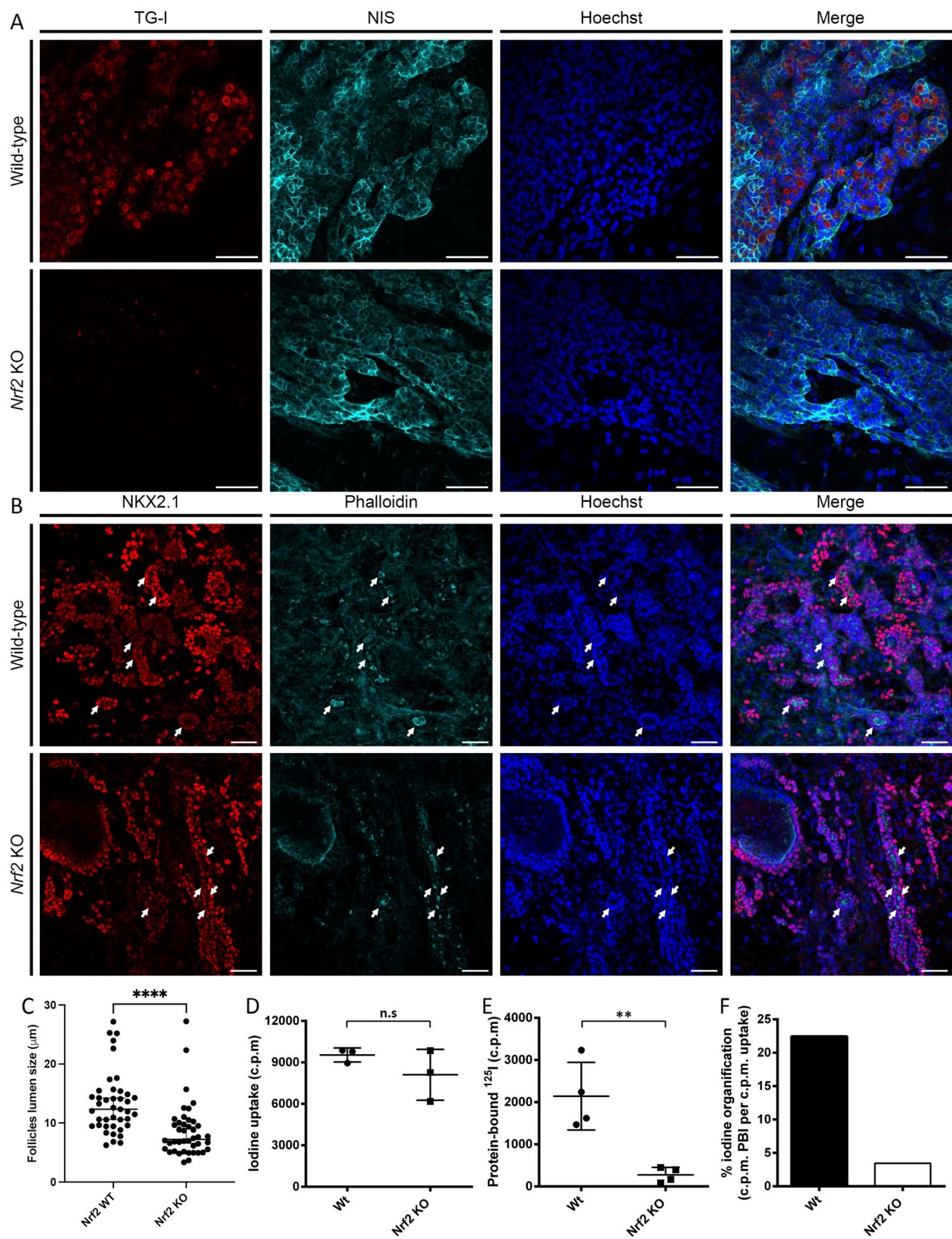

**Figure 5. Loss of Nrf2 impairs thyroid function in mESC-derived thyroid follicles.**
Characterization of mESC-derived thyroid organoids' function reveals the critical need for functional Nrf2 for thyroid hormone synthesis. **(A)** Confocal imaging after immunofluorescence of WT and Nrf2 KO organoids after 22 d of differentiation. Samples were marked for Tg-I (red) and Nis (cyan). Nuclei were labeled using Hoechst (blue). Scale bars: 100 $\mu$M. **(B)** Confocal imaging after immunofluorescence of WT and Nrf2 KO organoids after 22 d of differentiation. Samples were marked for Nkx2.1

displayed the presence of Stat1/Stat5 TF binding motifs in the 5-kb upstream promoter region (Fig 6H).

To further investigate whether AP-1 up-regulation in Nrf2 KO–derived thyroid organoids has a compensatory effect or whether it contributes to the phenotype, we performed AP-1 inhibition during the thyroid differentiation protocol. Nrf2 KO cells differentiated with the SR11302 inhibitor (from days 7 to 22) did not show recovery in Tg mRNA (Fig S7A), even exhibiting a further down-regulation at the higher dose (10 $\mu$M), and Tg protein and Tg-I production were also not affected (Fig S7B). This suggests that AP-1 up-regulation upon loss of Nrf2 might be a compensatory mechanism; however, because of the lack of Tg expression, which is under a direct effect of Nrf2, the functionality is not recovered.

## Discussion

In this study, we explored the role of the Nrf2/nrf2a TF during development and functional maturation of the thyroid gland. Using zebrafish embryos, we ruled out the role of Nrf2 on early thyroid organogenesis, whereas in mESC-derived thyroid organoids, it plays a role in Tg production and consequently in folliculogenesis. Importantly, in both models, we show that the loss of Nrf2/nrf2a affects thyroid functional maturation. Using a nonmammalian in vivo model and a mammalian in vitro model, we describe the similarities and the differences in Nrf2's role in thyroid late development (summarized in Fig 7).

### Lack of Nrf2/nrf2a in zebrafish embryos and mESC-derived thyrocytes induces dyshormonogenesis

Using zebrafish, we identified nrf2a as an actor of thyroid functional maturation. Upon nrf2a loss-of-function, early thyroid development occurs normally. However, functional maturation of the thyroid gland appears impaired and characterized by a hyperplastic nonfunctional thyroid gland, suggestive of a thyroid dyshormonogenesis phenotype.

Under physiological conditions, thyroid function is tightly regulated by the hypothalamus–pituitary axis via the TSH signaling pathway (Vassart & Dumont, 1992; Carvalho & Dupuy, 2017). Consistent with the lack of T.H., nrf2a mutants showed increased mRNA levels of tshβ, slc5a5, and tpo. Distinct from the hypothyroid zebrafish (Opitz et al, 2011; Giusti et al, 2020), nrf2a$^{-/-}$ embryos did not display an increase in tg mRNA and protein expression. Despite that the expression of key maturation factors such as tg, slc5a5, duox, and tpo is maintained in nrf2a$^{-/-}$ embryos, a disruption in duox and/or tpo enzymes and a consequent impairment in tg iodination cannot be ruled out (Giusti et al, 2020). Duox enzymes are known for tightly controlling $H_2O_2$ production, an essential factor for T.H. synthesis (Carvalho & Dupuy, 2017). Additional

studies are needed to assess a possible secondary effect of loss of functional nrf2a in promoting duox/tpo activity dysregulation and the iodination defect leading to hypothyroidism.

Our findings in zebrafish embryos suggest that the role of nrf2a goes beyond the expression control of metabolic and oxidative stress response genes and reinforces a published report on the role of Nrf2 in adult mouse thyroid physiology (Ziros et al, 2018). In this study, Nrf2 KO mice display a euthyroid state characterized by normal levels of Tsh, reduced basal levels of Tg, and increased Tg-I. To assess the role of Nrf2 during mammalian development, we used our previously published mESC-derived thyroid organoid model (Antonica et al, 2012; Romitti et al, 2021). Nrf2 KO mESC-derived organoids showed a higher proportion of NKX2.1+ cells than Nrf2 WT cells. However, the mechanism behind this phenomenon is unclear. Nrf2 has been proposed to promote mesendoderm cell fate in human and mouse embryonic stem cells (Jang et al, 2016; Dai et al, 2020), which could explain favoring of nonmesodermal cell lineages in organoids upon loss of Nrf2 expression. Other studies highlighted the role of Nrf2 in controlling the stemness and quiescence of stem and progenitor cells (Tsai et al, 2013; Murakami et al, 2017; Dai et al, 2020). In this study, no clear differences were observed between Nrf2 WT and KO mESC pluripotency and proliferation; however, clonal differences cannot be ruled out. mESC-derived thyroid organoids showed a dyshormonogenesis phenotype similar to the one seen in zebrafish, but distinct from the Nrf2$^{-/-}$ adult mouse thyroid model (Ziros et al, 2018). Despite the similarities described above, significant differences could be observed between the zebrafish thyroid gland, mESC-derived thyroid organoids, and the in vivo mouse model upon loss of Nrf2/nrf2a expression (Fig 7). The mESC Nrf2 KO–derived thyrocytes displayed a significant reduction in Tg expression, whereas the nrf2aΔ5 mutant zebrafish showed a clear reduction in tg expression only under a stress condition (PTU treatment). Firstly, in rodents, Nrf2 was shown to directly control Tg expression through ARE sequences present in the Tg promoter (Ziros et al, 2018). Our ATACseq analysis further confirmed these findings by showing a reduction in chromatin accessibility at Tg loci upon loss of Nrf2 expression. Interestingly, we did not detect any canonical ARE sequence (Fig 6C) in the zebrafish tg promoter, which could explain the difference in Tg regulation between the two models. Another possible explanation could be the availability of Tsh as a signaling molecule. Conversely to our zebrafish model, where thyroid function is regulated by the hypothalamus–pituitary–thyroid axis, mESC-derived thyroid organoids are not stimulated with Tsh during the differentiation protocol. Instead, we used cAMP (Romitti et al, 2021) as it is known for mediating most of Tsh signaling effects, such as thyrocyte proliferation and differentiation, hormone biosynthesis, and release (Ortiga-Carvalho et al, 2016). The role of Tsh signaling in controlling thyroglobulin expression is still a matter of debate. In zebrafish embryos, inhibition of Tsh signaling induced a

---

(red) and phalloidin (cyan). Nuclei were labeled using Hoechst (Blue). Follicles are indicated by arrows. Scale bars: 100 $\mu$m. **(C)** Thyroid organoid follicle size assessment after 22 d of differentiation. For each condition, samples were labeled using phalloidin staining and 40 follicles were analyzed using Zen software. The median follicle size is 12.34 and 7.22 $\mu$m for Nrf2 Wt and Nrf2 KO organoid, respectively. Statistical analysis was performed using the Mann–Whitney test. The asterisks denote significant differences between conditions: ****$P$ < 0.0001. **(D, E, F)** Functional organification assay revealed an equivalent ability of Nrf2 KO organoids to $^{125}$I uptake (D) but reduced protein-bound $^{125}$I fraction in Nrf2 KO organoids (E), which results in a lower percentage of cells with capacity of $^{125}$I organification (F). Statistical analysis was performed using a $t$ test. The asterisks denote significant differences between conditions: **$P$ < 0.01. "n.s", no significant difference.

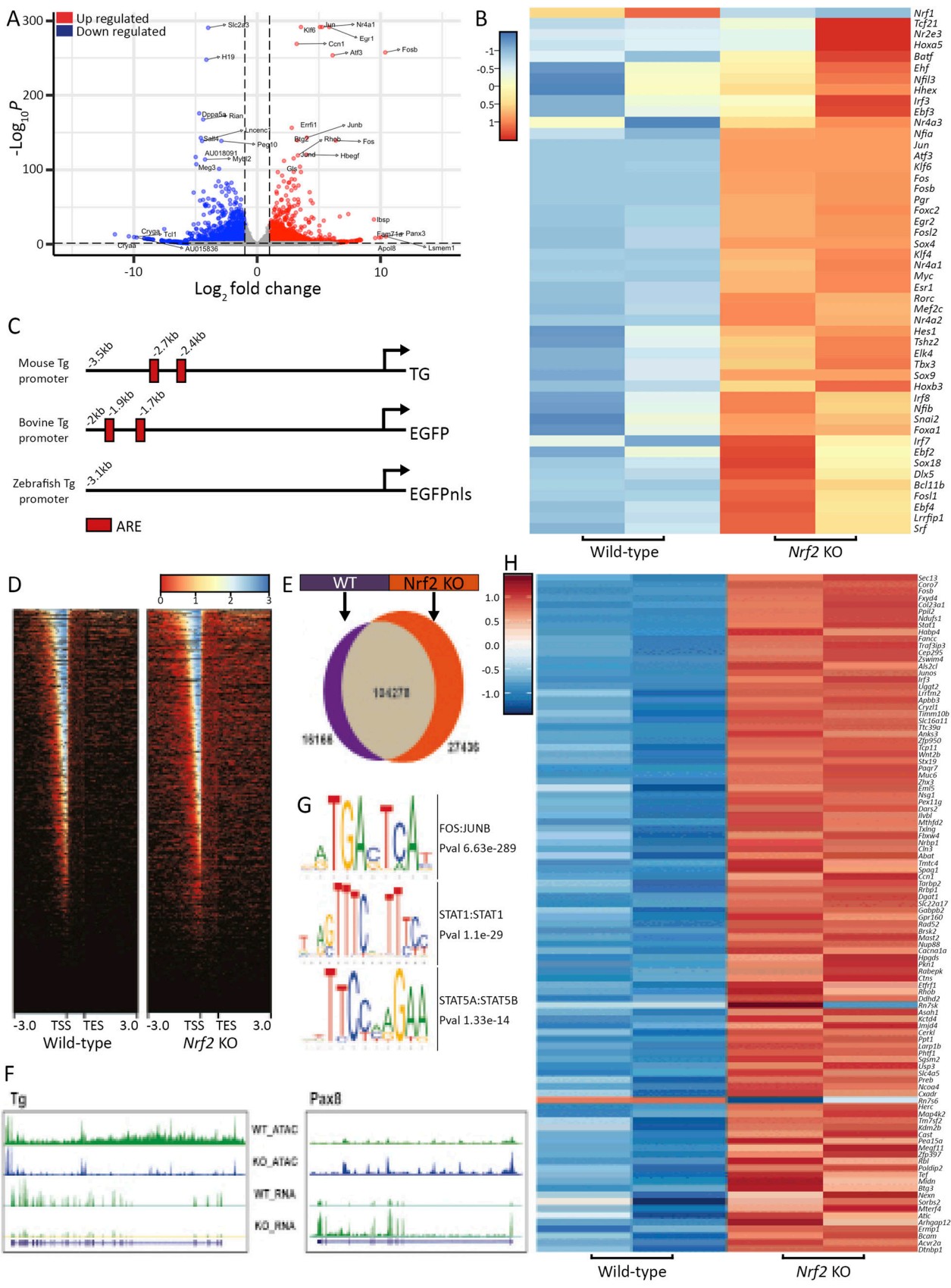

reduction of Tg expression (Opitz et al, 2011), although not as striking as for *slc5a5* and *tpo*. Although some studies have demonstrated the TSH-independent onset of Tg expression in mice (Marians et al, 2002; Postiglione et al, 2002; Christophe, 2004), others have suggested that it could play a role in maintaining Tg expression, in adult rat and dog thyroids (van Heuverswyn et al, 1984; Gérard et al, 1989; Christophe, 2004). In our study, Nrf2 WT and KO thyroid organoids differentiated with TSH instead of cAMP showed the same down-regulation in Tg mRNA expression, suggesting that the effect is mainly driven by the lack of Nrf2 biding to the ARE sequence of Tg promoter.

Morphologically, mESC Nrf2 KO–derived organoids exhibited increased variability in follicular organization, with generally smaller follicles compared with controls, a phenotype that was not observed in zebrafish embryos. Thyroid folliculogenesis is a poorly understood process, and recent studies suggest that thyroid follicle formation occurs in two steps. The first step is endothelial cell–dependent basal laminal formation around unpolarized thyroid cells (Villacorte et al, 2016; Pierreux, 2021). The second step is the formation of a microlumen through the migration of thyroglobulin-filled microvesicles to the opposite side of the basal lamina (Gonay et al, 2021). Although follicular lumen expansion is a thyroglobulin-independent process (Liang et al, 2018), the initial microlumen formation may be thyroglobulin-dependent, although additional studies in thyroid organoids lacking Nrf2 are necessary to confirm this hypothesis. In adult mice lacking Nrf2 expression, thyroid folliculogenesis seems to be preserved, which explains the normal T.H. production (Ziros et al, 2018). Our results show variability in folliculogenesis and T.H. production in early thyroid organoid development but do not exclude that throughout later stages, Tg accumulation could occur, leading to proper follicle formation and consequent T4 production.

In zebrafish, only few studies have described thyroid folliculogenesis and showed that soon after budding out of the endoderm, thyroid cells are already organized as a multicellular follicle-like structure (Alt et al, 2006; Porazzi et al, 2009; Opitz et al, 2012). In addition, the zebrafish thyroid gland can produce thyroid hormones while proliferating and migrating toward its final location, which greatly differs from mammalian thyroid development (Opitz et al, 2011, 2012; Nilsson & Fagman, 2017). Evolutionarily, this difference is thought to be caused by the earlier need for thyroid hormones in zebrafish embryos compared with the mouse embryos. Indeed, as soon as it hatches (2–3 dpf), the zebrafish embryos start to feed on prey, which requires the maturation of its gastrointestinal apparatus, the inflation of its swim bladder, and the maturation of its craniofacial cartilages. All these processes are controlled by thyroid hormones (Liu & Chan, 2002; Bagci et al, 2015; Campinho, 2019). Although it is currently admitted, yet not proven, that zebrafish and mouse folliculogenesis follow similar mechanisms, our data in thyroid organoids support the findings, suggesting that in mammals, folliculogenesis is a Tg-dependent process. In zebrafish, up to date, there are no data suggesting that the folliculogenesis could follow a different process.

## Loss of *Nrf2* affects the transcriptome and chromatin landscape of mESC-derived thyrocytes

Analysis of the transcriptome of mESC *Nrf2* KO–derived thyrocytes revealed the up-regulation of a TF network. Among those TFs, several are important effectors of the AP-1 network, such as *Atf*, *Jun*, and *Fos* family. AP-1 TFs play an important role in controlling cell proliferation, survival, and death (Karin et al, 1997), whereas it can be stimulated by growth factors, proinflammatory cytokines, and UV radiation (Karin, 1995), all of which display enriched signatures in *Nrf2* KO cells. JunB and Nrf2 were shown to form a heterodimer to modulate ARE-driven genes (Venugopal & Jaiswal, 1998), yet the regulatory link between Nrf2 and JunB remains unexplored in thyrocytes. Chromatin landscape analysis provided insights into the mechanism underlying the changes in the transcriptome. Enrichment of Fos/Jun and Stats binding motifs was found in *Nrf2* KO–derived thyrocytes. These motifs were detected within the promoter sequence of most differentially expressed genes, suggesting that in part, the gene expression changes could be caused by the up-regulation of both Fos and Stat TF families. In addition, three potential ARE sites within 1 kb upstream of the transcription start site of *Nrf2*, *Fosb*, and *Stat5a* were identified. Interestingly, Nrf2 and AP-1 proteins are all basic leucine zipper (bZip) TFs of similar structure (Novotny et al, 1998; Sykiotis & Bohmann, 2010; Babu et al, 2013). The role and regulation of Nrf2 and AP-1 proteins also significantly overlap regulating each other at several levels, including transcriptionally (Xanthoudakis et al, 1992; Venugopal & Jaiswal, 1998; Kim et al, 2003; Iwasaki et al, 2006). Another important aspect is that although Nrf2 binds to ARE and AP-1 to TRE sites, they often overlap with AP-1 being embedded into ARE (Reichard & Petersen, 2004; Zolotukhin et al, 2013). These tight relations suggest shared feedforward and feedback circuits between NRF2 and AP-1 factors contributing to their functioning. Indeed, inhibition of AP-1 was shown not to be sufficient to recover Tg and Tg-I levels, suggesting that AP-1 up-regulation upon loss of Nrf2 has not a causal effect but rather as a compensatory mechanism. However, because Tg expression regulation is under a direct effect of Nrf2, AP-1 overexpression per se cannot promote thyroid functionality.

In summary, our data suggest an evolutionarily conserved importance of nrf2/nrf2a in the later stages of thyroid development in

---

**Figure 6. Transcriptomic and chromatin accessibility profile comparison between mESC-derived Nrf2 WT and KO organoids.**
**(A)** Volcano plot showing the distribution of the down- and up-regulated genes among Nrf2⁻/⁻ cells compared with the control condition (WT). **(B)** Heatmap of normalized bulk RNA-seq expression of up-regulated transcription factor in *Nrf2* KO–derived thyrocytes. **(C)** Schematic representation of mouse, bovine, and zebrafish Tg promoter based on available sequence (mouse promoter) and sequences used as reporter drivers (bovine and zebrafish promoters). ARE sequences are represented as red boxes, and their relative position to the end of the promoters is given. **(D)** Heatmaps show profiling of chromatin accessibility around gene transcription start sites (TSS) in *Nrf2* WT (left panel)– and KO (right panel)–derived organoids. **(E)** Venn diagram representing the differentially represented peak in *Nrf2* WT (purple)– and *Nrf2* KO (orange)–derived organoids. **(F)** RNA and ATACseq browser track of *Tg* (left panel) and *Pax8* (right panel) showing reduced *Tg* and increased Pax8 mRNA levels. These differences in transcript levels correlated with differential chromatin accessibility peaks. **(G)** Transcription factor binding motif enrichment in Nrf2 KO–derived organoids. **(H)** Heatmap of normalized bulk RNA-seq expression displaying either FOS:JUNB or STAT54:STAT5B binding motifs in the 5 kb upstream of their TSS.

|  | Zebafish *nrf2a* KO | *Nrf2* KO derived thyrocytes | *Nrf2* KO mice |
|---|---|---|---|
| Nrf2 functionality | ↓↓↓ | ↓↓↓ | ↓↓↓ |
| Tg mRNA | Non-significant reduction | ↓ | ↓ |
| Tg Protein | Unaffected | ↓ | ↓ |
| ARE in Tg promoter | No | Yes | Yes |
| Iodinated Tg | ↓↓↓ | ↓↓↓ | ↑ |
| T4 | ↓↓↓ | ↓↓↓ | slight increase upon standard condition |
| Folliculogenesis | Unaffected | Poorly efficient | Unaffected |

**Figure 7. Summary of the observation in zebrafish *nrf2a* KO and *Nrf2* KO mESC-derived thyrocytes.**
Arrows indicate changes compared with either *nrf2a* WT zebrafish embryos or *Nrf2* WT mESC-derived thyroid organoids. The third column is summarizing observation made by Ziros and colleagues in the Nrf2 KO mouse model (adapted from Ziros et al [2018]). ARE, antioxidant response element.

mammals and other vertebrates. Using a combination of in vivo and in vitro models, we characterize the effect of *Nrf2*/*nrf2a* loss-of-function on thyroid organogenesis at the morphological and transcriptomic levels. Although our study highlighted a common dyshormonogenesis phenotype, its main driver appears to be distinct in *nrf2a* KO zebrafish embryos and in mESC *Nrf2* KO–derived thyrocytes. Although a strong reduction of Tg expression was observed in our in vitro model, the Tg protein appeared unchanged in vivo. Moreover, upon lack of functional *Nrf2*, zebrafish embryos display a normal folliculogenesis process, which is variable in mESC-derived organoids. The molecular mechanism leading to the Nrf2 loss phenotype is not fully understood, but our data suggest that Nrf2/AP-1 tight transcriptional regulation plays an important role in thyroid maturation and function.

## Materials and Methods

### Animal husbandry

Zebrafish embryos were collected by natural spawning of adult individuals; embryos and larvae were raised at 28.5°C under standard conditions (Westerfield, 2000) and staged according to hours (hpf) or days post-fertilization (dpf) as described (Kimmel et al, 1995). Embryos were collected and raised in 90-mm petri dishes containing 25 ml of embryo medium. At 1 dpf, all embryos were dechorionated by a 20-min treatment with 0.6 mg/ml pronase (Sigma-Aldrich). Screening and live imaging of embryos and larvae were performed after anesthesia using 0.02% tricaine (Sigma-Aldrich). The following zebrafish lines were used in this study: WT AB strain, Tg(*tg:nlsEGFP*)[ulb4] (Trubiroha et al, 2018), *nrf2aΔ5* mutant generated by CRISPR/Cas9-based mutagenesis (this study), and Tg(*tg:nrf2a_T2A_mKO2nls*) (this study). All zebrafish husbandry and all experiments were performed under standard conditions following institutional (Université Libre de Bruxelles [ULB]) and national ethical and animal welfare guidelines and regulations. All experimental procedures were approved by the Ethical Committee for Animal Welfare (CEBEA) from the Université Libre de Bruxelles (protocol 578 N).

### sgRNA design and synthesis

The design and generation of sgRNA used in this study were performed as previously described (Trubiroha et al, 2018).

Briefly, the Sequence Scan for CRISPR software (http://crispr.dfci.harvard.edu/SSC/) was used to identify sgRNA sequences with predicted high on-target activity (Xu et al, 2015). DNA templates for sgRNA synthesis were produced using the PCR-based short-oligo method as previously described (Talbot & Amacher, 2014). DNA templates for *in* vitro transcription of sgRNA were generated by annealing a scaffold oligonucleotide sequence: AAAGCACCGACTCGGTGCCACTTTTTCAAGTTGATAACGGACTAGCCTTATTTTAACTTGCTATTTCTAGCTCTAAAAC, with gene-specific oligonucleotides containing the SP6 promoter sequence (GCGATTTAGGTGACACTATA), a 20-base target sequence (see Table S1), and a complementary sequence to the scaffold oligo (GTTTTAGAGCTAGAAATAG). PCR amplification was performed using Taq PCR Core Kit (QIAGEN). The reaction mix contained 20 nM scaffold oligo, 20 nM gene-specific oligo, and 260 nM of each universal flanking primer (forward: GCGATTTAGGTGACACTATA, reverse: AAAGCACCGACTCGGTGCCAC). Purification of the PCR products was performed using a MinElute Reaction Cleanup kit (QIAGEN) followed by a phenol–chloroform–isoamyl alcohol extraction. sgRNAs were synthesized in vitro from purified PCR products using the SP6 RNA polymerase (NEB). 20 $\mu$l reactions contained 1 $\mu$g DNA template, 1x reaction buffer, 2.5 mM of each rNTP (Roche), 40 U RNase inhibitor (Thermo Fisher Scientific), and 6 U Large Fragment of DNA Polymerase I (Thermo Fisher Scientific). After preincubation for 15 min at RT, 20 U SP6 RNA polymerase was added and reactions were run overnight at 40°C. After treatment with 3 U DNase I (AmpGrade; Thermo Fisher Scientific), sgRNAs were purified using High Pure PCR Cleanup Micro Kit (Roche). The concentration of sgRNAs was measured by NanoDrop (Thermo Fisher Scientific), and integrity was checked by gel electrophoresis. Aliquots of sgRNA solutions were stored at –80°C until use.

### sgRNA and Cas9 injection

Injection mix containing Cas9 protein (PNA Bio; 100 ng/$\mu$l final concentration), sgRNA (80 ng/$\mu$l final concentration), 200 mM KCl, and up to 0.15% phenol red (Sigma-Aldrich) was injected into one-cell-stage embryos. Approximately 3 nl of active sgRNA-Cas9 ribonucleoprotein complex was injected in each one-cell-stage Tg(tg:nlsEGFP) embryo. For each sgRNA, at least three independent injection experiments were performed with spawns from at least three different founder fish.

## Phenotypic analysis of F0 crispants and stable mutant embryos

After anesthesia using 0.02% tricaine, thyroid development was monitored in live zebrafish at 55 hpf, 80 hpf, and 6 dpf by visual inspection using a M165 FC fluorescence stereomicroscope (Leica). For F0 crispant analysis, all injected and noninjected embryos were inspected for gross developmental defects, deviations in size, shape, or location of the thyroid, and for the overall intensity of the fluorescent reporter signal. For imaging of the thyroid phenotypes, embryos and larvae were immobilized in 1% low-melting-point agarose (Lonza) containing 0.016% tricaine and positioned in Fluoro-Dish glass-bottom dishes (World Precision Instruments). Imaging was performed using a DMI600B epifluorescence microscope equipped with a DFC365FX camera and LAS AF Lite software (Leica). If indicated, confocal live imaging was performed with a LSM 780 confocal microscope (Zeiss) using Zen 2010 D software (Zeiss).

## Genotyping of zebrafish *nrf2aΔ5* mutants

Genomic DNA was extracted by a 10-min lysis treatment with 50 mM NaOH at 95°C either from adult tailfin clips or from whole larvae following. Subsequently, the lysate was neutralized using 0.5 M Tris (pH 8.0) and used as template PCRs using primers spanning the mutated site. The following primer sequences were used: forward 5′-TGATCATCAATCTGCCCGTA-3′ and reverse 5′-TCTCTTTCAGGTTGC TGCTG-3′. PCR products were then analyzed by Sanger sequencing to identify WT, heterozygous, and homozygous carriers.

## Phenylthiourea treatment of zebrafish embryo

Zebrafish embryos and larvae were treated with phenylthiourea (PTU; Sigma-Aldrich) from 24 hpf onward to induce hypothyroid conditions (Elsalini & Rohr, 2003; Opitz et al, 2011). A stock solution of 100 mg/l PTU was diluted in embryo medium to make experimental treatment solutions of 30 mg/l PTU. A control group was maintained in the embryo medium. Embryos and larvae were treated in 90-mm petri dishes containing 25 ml of treatment solution, and solutions were renewed every other day.

## WIF

144 hpf larvae were fixed in 4% PFA (pH 7.3) overnight at 4°C and stored in PBS containing 0.1% Tween-20 at 4°C until use. WIF experiments were performed as previously described (Opitz et al, 2011) using the following antibodies: goat anti-T4 polyclonal antibody (1:1,000; Biorbyt), chicken anti-GFP polyclonal antibody (1:1,000; Abcam), mouse anti-iodinated thyroglobulin polyclonal antibody (1:1,000; Dako), Cy3-conjugated donkey anti-goat IgG antibody (1:250; Jackson ImmunoResearch), Alexa Fluor 488–conjugated goat anti-chicken IgG antibody (1:250; Invitrogen), and Cy5-conjugated donkey anti-mouse IgG antibody (1:250, Jackson ImmunoResearch). After immunostaining, samples were postfixed in 4% PFA and gradually transferred to 100% glycerol for conservation at 4°C. Phenotypic analysis was performed using a M165 FC fluorescence stereomicroscope (Leica). Whole-mount imaging of stained specimens was performed either with a

DMI600B epifluorescence microscope (Leica) equipped with a DFC365FX camera or, if stated, using a LSM 780 confocal microscope (Zeiss).

## Quantitative analysis of T4 and Tg-I fluorescent signal intensities

Quantification of the fluorescent signal was performed as previously described (Thienpont et al, 2011), and whole-mount imaging of the thyroid region was performed on stained samples using a DMI600B epifluorescence microscope. Using LAS AF imaging software (Leica), pixel sum values were obtained for each thyroid follicular lumen (region of interest 1 to X, ROI 1-X) and adjacent nonfluorescent regions (ROI-bg, background). For each embryo, the intensity of each ROI 1-X was corrected by subtracting the intensity of ROI-bg. The sum of each individual background-corrected fluorescence signal intensity of a single embryo constitutes the measurement of either its T4 or Tg-I content. Values from individual fish were used in comparisons between treatment or mutant groups and controls.

## Generation of the IscE1 meganuclease-based Tg(*tg: nrf2a_T2A_mKO2nls*) transgenic zebrafish line

The Gateway cloning technology–based Tol2kit (Kwan et al, 2007) was used to generate a rescue construct in which the functional *nrf2a* coding sequence fused to a nuclear-localized mKO2 fluorescent protein with a viral T2A linker peptide (Kim et al, 2011) is expressed under the control of a fragment of the zebrafish *tg* gene promoter previously shown to allow thyroid-specific transgene expression (Opitz et al, 2012). Generation of the plasmid was essentially performed as previously described (Trubiroha et al, 2018). The full nrf2a coding sequence fused to the mKO2 protein via the T2A peptide was synthesized by Genewiz Company and was subsequently cloned into a pME vector to create a pME-*nrf2a*_T2A_mKO2nls vector. Assembly of the final Tol2Tg(*tg:nrf2a_T2A_mKO2nls*) vector was performed using a LR reaction (Villefranc et al, 2007).

Capped mRNA encoding for transposase was generated by in vitro transcription of the NotI-linearized pCS-zT2TP plasmid (Kawakami, 2004) using the mMessage mMachine SP6 kit (Ambion). Transgenic embryos were generated by the co-injection of Tol2Tg(*tg:nrf2a_- T2A_mKO2nls*) vector (25 ng/µl) and Tol2 transposase mRNA (35 ng/µl) in one-cell-stage *nrf2aΔ5* heterozygous embryos. Injected embryos were screened at 55 and 80 hpf for mKO2 expression in thyroid cells, and only F0 fish presenting thyroid-specific mKO2 expression were raised to adulthood to identify potential germline transmitters. F0 founders with germline transmission were outcrossed with WT fish to generate stable F1 transgenic Tg(*tg:nrf2a_T2A_mKO2nls*) animals. Transgenic F1 embryos were raised to adulthood and genotyped for the presence of the *nrf2aΔ5* mutation. A Tg(*tg:nrf2a_T2A_mKO2nls*) line was maintained in a WT and in a *nrf2aΔ5* mutant genetic background.

## Generation of the A2lox_Nkx2-1_Pax8_Nrf2$^{-/-}$ mouse ESC line using CRISPR/Cas9

SgRNAs targeting mouse Nrf2 coding sequence were selected using the prediction algorithm CRISPR, as described above. To compare

the effect of Nrf2 knockout in mESC-derived thyroid with zebrafish nrf2aΔ5 mutants, the selected gRNA targets the same genomic region chosen for zebrafish, the DNA binding and transactivation of the *Nrf2* gene. The most suitable sgRNA was chosen and checked for possible off-targets. Initially, the gRNA sequence (GTGTGAGAT GAGCCTCTAAG) was cloned inside the pU–BbsI-T2A-Cas9-BFP plasmid (64323; Addgene). Then, the A2Lox.Cre_TRE-Nkx2-1/ Pax8_Tg-EGFP mouse ESCs (Romitti et al, 2021) were transfected using Lipofectamine 3000 reagent (Invitrogen), and after 24 h, BFP-expressing cells were sorted, clonal-seeded, expanded and analyzed by PCR, and sequenced. For that, genomic DNA was extracted from individual mESC clones using DNeasy Blood and Tissue Kit from QIAGEN. PCR was performed using 100 ng of genomic DNA, and Q5 High-Fidelity Taq Polymerase (New England Biolabs) was used according to the manufacturer's instructions. Primer sets (Fw: 5′-TCCTATGCGTGAATCCCAAT-3′ and Rv: 5′-TAAGTGGCCCAAGTCTTGCT-3′) were used to amplify the CRISPR/Cas9 target region. PCR products were cloned into the TOPO-Blunt plasmid using Zero Blunt PCR Cloning Kit (Thermo Fisher Scientific) and sequenced using the Sanger method. Clones were chosen based on the identification of frameshift mutations in the *Nrf2* gene. For thyroid differentiation, two different clones were selected and validated according to the maintenance of pluripotency, tested by spontaneous differentiation into the three germ layers and induction of Nkx2-1-Pax8 transgenes under Dox stimulation.

## mESC_Nkx2-1_Pax8_Nrf2$^{-/-}$ culture and thyroid differentiation

Modified mESCs were cultured and differentiated as previously described (Antonica et al, 2012, 2017). Briefly, cells were cultured on irradiated mouse embryonic fibroblast feeder layer using a mESC medium (Antonica et al, 2012, 2017). For thyroid differentiation induction, Nkx2-1_Pax8_Nrf2$^{-/-}$ and Nkx2-1_Pax8_Nrf2$^{wt/wt}$ (control) mES cells were cultured for 4 d in suspension using the hanging drop method to generate embryoid bodies (EBs). After 4 d, generated EBs were collected and embedded in 50 $\mu$l (around 40 EBs) of growth factor–reduced Matrigel (BD Biosciences), plated in 12-well plates, and treated for 3 d with 1 mg/ml doxycycline (Sigma-Aldrich), to stimulate Nkx2-1 and Pax8 transgene expression. After Dox withdrawal, cells were treated for 2 wk with 300 $\mu$M 8-br-cAMP (Biolog Inc.) or 1 mU/ml rhTSH (Genzyme) (Romitti et al, 2021). Samples were collected at distinct time points for RNA expression and histological analysis.

## Quantitative PCR (qRT-PCR) analysis

qRT-PCR was performed on cDNA generated from thyroid organoids derived from *Nrf2* WT and KO cells at differentiation day 7 and 22 or from whole zebrafish embryos at 6 d of age. Total RNA was extracted from samples using the RNeasy micro kit (QIAGEN) according to the manufacturer's instructions. cDNA was generated by reverse transcription using the Superscript II kit (Invitrogen). qRT-PCR was performed in triplicates using Takyon (Eurogentec) and CFX Connect Real-Time System (Bio-Rad). Results are presented as linearized values normalized to the housekeeping gene, B2-microglobulin, and the indicated reference value (2-DDct). The gene expression profile was obtained from at least three independent samples and compared with their respective control (Nkx2-1_Pax8_Nrf2$^{wt/wt}$ mESC line or WT zebrafish embryos). Primer sequences are listed in Table S1.

## Flow cytometry intracellular immunostaining

Nrf2$^{wt/wt}$- and Nrf2$^{-/-}$-derived cells were collected at day 22 (cAMP condition) and prepared for flow cytometry immunostaining as follows: Matrigel-embedded organoids were digested using a HBSS solution containing 10 U/ml dispase II (Roche) and 125 U/ml of collagenase type IV (Gibco, Thermo Fisher Scientific) for 30–60 min at 37°C; then, enzymes were inactivated using 10% FBS followed by incubation with TripLE Express solution (Thermo Fisher Scientific) for 10–15 min, at 37°C, to dissociate remaining structures (including thyroid follicles) and finally obtain single-cell suspension. After inactivation by the addition of differentiation medium, cells were centrifuged, rinsed with PBS, and fixed in 1.6% PFA solution in PBS for 15 min at RT, followed by cell permeabilization with 0.1% Triton solution in PBS for 15 min at 4°C under agitation. After centrifugation, cells were blocked using 4% horse serum (HS) and 0.5% Tween-20/PBS blocking solution for 10 min (4°C under agitation). Primary anti-rabbit Nkx2-1 antibody (1:100) was diluted in blocking solution and samples incubated for 30 min (4°C under agitation). Cells were then rinsed three times with washing solution (0.5% BSA and 0.5% Tween in PBS) and then incubated with Cy5-conjugated anti-rabbit antibody (1:300, in blocking solution) for 30 min (4°C under agitation). Data concerning bTg-GFP (endogenous expression) and Nkx2-1 expression were obtained and processed using a LSRFortessa X-20 flow cytometer and FACSDiva software (BD Biosciences), respectively. Unstained cells and isotype controls were included in all experiments. In addition, GFP+ cell quantification was used to estimate the proportion of Tg+ cells generated by Nrf2$^{wt/wt}$ and Nrf2$^{-/-}$ cells.

## Immunofluorescence

Protein immunodetection experiments were performed as previously described (Antonica et al, 2012; Romitti et al, 2021). Briefly, organoids were fixed in 4% paraformaldehyde (Sigma-Aldrich) for 1 h at RT, washed three times in PBS, and blocked in a solution containing 3% BSA (Sigma-Aldrich), 5% horse serum (Invitrogen), and 0.3% Triton X-100 in PBS (Sigma-Aldrich) for 30 min at RT. The primary and secondary antibodies were diluted in a PBS solution containing 3% BSA, 1% horse serum, and 0.1% Triton X-100. The following primary and secondary antibodies were used in this study: mouse anti-iodinated thyroglobulin (1/1,000; generous gift from C. Ris-Stalpers), rabbit anti-Nkx2.1 (1/750; BioPAT), mouse anti-Nis (1/300; Sigma-Aldrich), mouse anti-thyroglobulin (1/300; Sigma-Aldrich), rabbit anti-thyroglobulin (1/1,000; Dako), Alexa Fluor 647–conjugated phalloidin antibody (1:100, Invitrogen), Cy3-conjugated donkey anti-goat IgG antibody (1:250; Jackson ImmunoResearch), Alexa Fluor 488–conjugated goat anti-chicken IgG antibody (1:250; Invitrogen), and Cy5-conjugated donkey anti-mouse IgG antibody (1:250; Jackson ImmunoResearch). Coverslips containing the stained organoids were mounted with Glycergel (Dako), and samples were imaged on Leica DMi6000 or Zeiss LSM 780 confocal microscope.

## Iodide organification assay

Nrf2$^{wt/wt}$- and Nrf2$^{-/-}$-derived organoids (cAMP-treated) were tested for the ability of iodide uptake and organification at differentiation day 22. Matrigel drops containing approximately the same number of organoids were initially washed with HBSS and incubated with 1 ml of a HBSS solution containing $^{125}$I (1,000,000 c.p.m./ml; PerkinElmer) and 100 nM of sodium iodide (NaI; Sigma-Aldrich) for 2 h at 37°C. After incubation, 1 ml of 4 mM methimazole (MMI) was added and cells were washed with ice-cold PBS. To dissociate the organoids from the MTG drops, cells were incubated with 0.1% trypsin (Invitrogen) and 1 mM EDTA (Invitrogen) in PBS for 15 min. For iodide uptake evaluation, cells were collected in polyester tubes and radioactivity was measured with a c-counter. After, proteins were precipitated using 1 mg of gamma-globulins (Sigma-Aldrich) and 2 ml 20% TCA followed by centrifugation at 710$g$ for 10 min, at 4°C, and the protein-bound $^{125}$I (PBI) was measured. Iodide organification was calculated by the iodide uptake/PBI ratio, and the values were expressed as a percentage. As control for iodide uptake and protein binding measurements, cells were also treated with 30 mM sodium perchlorate (Nis inhibitor; NaClO4; Sigma-Aldrich) and 2 mM methimazole (TPO inhibitor; MMI; Sigma-Aldrich), respectively. Experiments were performed using at least three independent replicates.

## Bulk RNA sequencing

Bulk RNA-seq was performed in Nrf2$^{wt/wt}$- and Nrf2$^{-/-}$-differentiated cells at day 22 of our differentiation protocol. Total RNA isolation was performed as described above (qRT-PCR section). The RNA concentration and quality were accessed using Bioanalyzer 2100 (Agilent) and RNA 6000 Nano Kit (Agilent). RNA integrity was preserved, and no genomic DNA contamination was detected. Illumina TruSeq RNA Library Prep Kit v2 was employed, as indicated by the manufacturer, resulting in high-quality indexed cDNA libraries, which were quantified using the Quant-iT Pico-Green kit (Life Sciences) and Infinite F200 Pro plate reader (Tecan); DNA fragment size distribution was examined on 2100 Bioanalyzer (Agilent) using the DNA 1000 kit (Agilent). The multiplexed libraries (10$\rho$M) were loaded on flow cells and sequenced on the HiSeq 1500 system (Illumina) in a high-output mode using HiSeq Cluster Kit v4 (Illumina). Roughly 10 million paired-end reads were obtained per sample. After removal of low-quality bases and Illumina adapter sequences using Trimmomatic software (Bolger et al, 2014), sequence reads were aligned against mouse reference genome (GRCm38/mm10) using STAR software with default parameters (Dobin et al, 2013). Raw counts were obtained using HTSeq software (Anders et al, 2015) using Ensembl genome annotation. Normalization, differential expression, and gene ontology analyses were performed using two biological replicates per sample, using website iDEP version 0.93 (Ge et al, 2018). Genes in which the expression levels were lower than 5 were filtered out.

## ATAC sequencing

ATAC-sequencing experiment was performed on biological replicates of Nrf2$^{wt/wt}$- and Nrf2$^{-/-}$-differentiated cells at day 22 of our differentiation protocol. About 50,000 Nkx2-1(mKO2+) cells were collected and immediately proceeded to sample preparation, based on the Omni-ATAC protocol (Corces et al, 2017). After centrifugation, cell pellets were resuspended in 50 $\mu$l of an ice-cold cell lysis buffer (0.1% Igepal, 0.1% Tween-20, and 0.01% digitonin in Omni-ATAC Resuspension Buffer). After 3 min, samples were centrifuged for 15 min at 800$g$, and subsequently, nuclei were resuspended in 50 $\mu$l of reaction buffer (2.5 $\mu$l Tn5 transposase, 22.5 $\mu$l TD buffer, both from Nextera DNA Sample Preparation Kit; 16.5 $\mu$l PBS, 0.5 $\mu$l 1% digitonin, 0.5 $\mu$l 10% Tween-20, and 5 $\mu$l H20). Tagmentation reaction was performed for 30 min at 37°C in a rocking plate (1,000 rpm). DNA was purified using the MinElute purification kit following the manufacturer's indications. DNA libraries were PCR-amplified, DNA quality–verified on 2100 Bioanalyzer (Agilent) using the DNA 1000 kit, and size-selected from 200 to 800 bp, following the manufacturer's recommendations. Mapping of ATACseq reads was performed using the Bowtie pipeline with standard parameters. Peak calling was performed using MACSv2 and downstream motif analysis using HOMER and MEME pipelines.

## Regulatory sequence analysis

The regulatory sequence for mouse Tg was downloaded from the EPDnew database (http://epd.vital-it.ch, Promoter ID: TG_1) (Dreos et al, 2017). Bovine and zebrafish Tg regulatory sequences used in this study are the same as the one used to develop the previously published reporter mESC and zebrafish lines (Antonica et al, 2012; Trubiroha et al, 2018). Identification of potential ARE sequences was done through the JASPAR database (http://jaspar.genereg.net) (Mathelier et al, 2016) using the ARE weight matrix MA0150.2. Only identified sequences with a $P$-value and a $Q$-value < 0.05 were considered for the analysis.

# Supplementary Information

# Acknowledgements

We thank members of the Costagliola laboratory for comments on the article and technical assistance and members of the IRIBHM fish facility for technical assistance. We thank J-M Vanderwinden from the Light Microscopy Facility and Christine Dubois from the FACS facility for technical assistance at ULB. Bulk RNA and ATAC sequencing were performed at the Brussels Interuniversity Genomics High Throughput Core (BRIGHTcore: www.brightcore.be). This work was supported by grants from the Belgian National Fund for Scientific Research (FNRS) (FRSM 3-4598-12, PDR T.0140.14; PDR T.0230.18, CDR-J.0145.16, GEQ U.G030.19) and the Fonds d'Encouragement à la Recherche de l'Université Libre de Bruxelles (FER-ULB), and it has received funding from the European Union's Horizon 2020 research and innovation program under grant agreement no. 825745. This work was also supported by a Privileged Partnership Grant between ULB and UNIL (to GP Sykiotis and S Costagliola) and the Swiss National Foundation Research Grants 310030_212558 and IZCOZ0_205415 (to GP Sykiotis). Work by SP Singh was supported by MISU funding from the FNRS (34772792 – SCHISM). P Gillotay, B Dassy, and B Haerlingen were supported by FRIA. M Romitti was

supported in part by the Brazilian National Council for Scientific and Technological Development (CNPq; Brazil), FNRS (Chargé de Recherche), and ULB. MP Shankar was supported by FNRS, and BF Fonseca by ULB. S Costagliola is Research Director at FNRS.

## Author Contributions

P Gillotay: conceptualization, formal analysis, methodology, data curation, and writing—original draft.

S Bangru: data curation, formal analysis, methodology, and validation.

B Dassy: methodology and data curation.

B Haerlingen: methodology and data curation.

MP Shankar: methodology and data curation.

BF Fonseca: conceptualization, formal analysis, methodology, and data curation.

PG Ziros: conceptualization, formal analysis, methodology, and data curation.

SP Singh: formal analysis, methodology, and data curation.

GP Sykiotis: conceptualization, data curation, formal analysis, supervision, validation, methodology, data curation, and writing—original draft, review, and editing.

M Romitti: conceptualization, data curation, formal analysis, supervision, funding acquisition, validation, methodology, and writing—original draft, review, and editing.

S Costagliola: conceptualization, data curation, formal analysis, supervision, funding acquisition, validation, and writing—original draft, review, and editing.

## Conflict of Interest Statement

The authors declare that they have no conflict of interest.

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
