## [Reviewer comments · Life Science Alliance]

The role of Nrf2 in thyroid maturation and hormone synthesis in vertebrate models

Pierre Gillotay, Sushant Bangru, Benjamin Dassy, Benoit Haerlingen, Meghna Shankar, Barbara Da Fonseca, Panos Ziros, Sumeet Singh, Gerasimos Sykiotis, Mirian Romitti, and Sabine Costagliola

DOI: <https://doi.org/10.26508/lsa.202603687>

Corresponding author(s): Sabine Costagliola, Université Libre de Bruxelles and Mirian Romitti, Université Libre de Bruxelles

Review Timeline:

Submission Date:	2026-03-09
Editorial Decision:	2026-04-06
Revision Received:	2026-04-23
Accepted:	2026-04-24

Scientific Editor: Tim Fessenden

Transaction Report:

Please note that the manuscript was reviewed at *Review Commons* and these reports were taken into account in the decision-making process at *Life Science Alliance*.

The logo for Review Commons, featuring the word "Review" in a blue serif font with a diagonal slash through the letter 'v', and the word "COMMONS" in a blue sans-serif font below it.

Reviews

Review #1

1. Evidence, reproducibility and clarity:

This paper deals with the role of the transcription factor Nrf2 in the thyroid gland of zebrafish and in a thyroid organoid model. The subject is of relevance since Nrf2 is known to control the cellular response to oxidative stress and the thyroid is an organ where protection of oxidative stress is of major relevance, given the production of reactive oxygen species during thyroid hormone biosynthesis.

The main result is that in Zebrafish (ZF) thyroid Nrf2 appears to be important for thyroid hormone formation since late stages ZF embryos deprived of NRF2 the levels of the thyroid hormone T4 and of its precursor, iodinated thyroglobulin(Tg), are very much decreased. However, there is no significant decrease of Thyroglobulin mRNA, albeit an impairment in the up-regulation of Tg by TSH could be observed. No effect is seen on the structure of the thyroid follicles and no developmental defect is observed, in contrast with the title of the paper.

Conversely, in the mouse thyroid organoid model the absence of Nrf2 results in an impressive decrease of Tg mRNA and in impaired formation of thyroid follicles.

The study is in most part elegant and technically impeccable. The data are well presented and organised as far as figure is concerned. However, much remains to be done on the interpretation and presentation of results. In addition, the text that has been put together sloppily, with many typing and punctuation mistakes and difficult to interpret sentences. A revision of typing and syntax is absolutely needed.

****Main concerns:****

1. The title of the paper needs to be changed. There is no evidence that there is a problem with thyroid development in ZF. The thyroid appears to be enlarged at the end of development, most likely as a consequence of increased TSH stimulation, but there is no developmental defect!
2. There is an evident contradiction, in ZF, between the marginal, if any, decrease of Tg mRNA and the impressive decrease in T4 and iodinated Tg. This, in my opinion, very interesting discrepancy, is never discussed. Perhaps the authors should look at the level of Tg protein. It is possible that there is an increased degradation or some negative translational control in absence of Nrf2? Alternatively, is it possible that there is a defect, yet unidentified, in the organification process? Certainly, to conclude, as the authors do in page 11, lines 236-237, that the defect in hormonogenesis depends on thyroglobulin production is, with the data presented, an unproven statement.
3. The data on transcriptional effect of NRf2 in the mouse ES cell system do not really add much. Their major effect is to contribute to a lengthy discussion that would really benefit of a substantial reduction.
4. More time should be devoted to explain the substantial differences between the three systems studied (two in this paper, one in a previous published by partly same authors), keeping in mind that studies in mice could be largely influenced by the genetic background.

2. Significance:

This paper is largely confirmatory of previous results obtained in a mouse Nrf2 KO model, whose main authors are also part of this study (Ziros et al. , 2018). A clarification of the molecular defect in hormone production in ZF could add the novelty that this study might need.

3. How much time do you estimate the authors will need to complete the suggested revisions:

Between 3 and 6 months

4. Review Commons values the work of reviewers and encourages them to get credit for their work. Select 'Yes' below to register your reviewing activity at Web of Science Reviewer Recognition Service (formerly Publons); note that the content of your review will not be visible on Web of Science.

Yes

Review #2

1. Evidence, reproducibility and clarity:

Summary

In this manuscript Gillotay et al investigate and further establish a role for Nrf2 in thyroid physiology. Importantly, a role of Nrf2 in thyroid development is investigated in the Tg(tg:nlsEGFP) zebrafish model system developed by this group. This permits detailed and also live tracking of zebrafish thyroid development. By a Crispr/Cas9 approach the authors establish a functional role for nrf2a in zebrafish thyroid development. Nrf2a loss-of-function in zebrafish leads to a hypothyroid phenotype that seems to recapitulate aspects of dyshormonogenesis with a slightly enlarged gland, increased tshb, strongly reduced T4 and iodinated TG (TG-I). Moreover, Tg synthesis is perturbed, only slightly during control conditions but pronouncedly during PTU challenge. By restoring nrf2a specifically in the zebrafish thyroid, the authors prove the phenotype to be cell autonomous to the thyroid.

The authors then turn to the model system of functional thyroid follicular maturation from mESC that they have previously described. In this system Nrf2a KO leads to loss of TG expression and what the authors describe as an inability to form follicular units. Even though Nis is expressed, iodine organification is impaired, likely due to defective Tg production.

Major comments

This is a nicely written manuscript with mostly convincing results. The authors demonstrate laudable scientific rigor by verifying that the genetic modifications indeed have the expected effect (that nrf2 in the model systems that they create is indeed non-functional) and by using a rescue strain for the zebrafish model. In my opinion the data and the methods are presented in such a way that they can be reproduced. Generally, the conclusions are well supported by the results presented.

The authors make a rather strong point of Nrf2 thyroid function as an evolutionary conserved mechanism and this might need some further underpinning.

Line 368. The authors state that their findings "...reinforce a recently published report on the role of Nrf2 in adult mouse thyroid physiology (Ziros et al., 2018)" and continue: "Although we did not analysed (sic!) the thyroid gland of adult nrf2a KO zebrafish"...".

Why were not adult zebrafish investigated? As some key results of Ziros et al on adult Nrf2 KO mice differ from those of the manuscript (e.g. TG iodination under non stressed conditions seems to be increased in mice but decreased in zebrafish) it would be highly desirable to know if the zebrafish phenotype is more similar to that of mice in later adulthood.

As I will elaborate on later on, to me a key finding seems to be that nrf2a might have unexpected "non-canonical functions" that do not immediately seem to pertain to its presumed major function as a key regulator of defense to oxidative stress. One of these is to be an impact on Tg production. The other one seems to be a possible role in folliculogenesis, even though this might in some way be related to the impact on Tg production. I do understand that the authors want to leverage from the elegant in vitro model of mESC-derived follicles that they have developed. However, the rationale of using the model system in this context is not entirely clear to me. In Ziros et al the effect of Nrf2 deficiency was studied in a murine global knockout model, in a thyroid specific knockout (even though it might be argued that by using a Pax8 Cre-driver Nrf2 was likely knocked out also in the kidneys and in some regions of the CNS) as well as in cultured rat thyroid cells. Is it the fact that the Pax8 Cre-driver is not entirely thyroid specific that prompted the authors to turn to the mESC-model, the possibility of studying thyroid cell autonomous mechanisms of folliculogenesis (without the possible impact of other tissues and the HPT-axis) by turning to this reductionistic model or other considerations? Even though the results from the mESC model certainly are of interest, the rationale needs to be better explained and the real potential of this model is perhaps not fully exploited. Specifically, the effect on folliculogenesis in the mESC system needs to be more carefully presented:

Line 294. Even though they might be discernible, it is difficult to really appreciate the occurrence of follicular lumina in Fig 4E WT structures. The authors consider this as a main finding (see discussion: "The most striking difference we observed between the two models was the absence of follicular organisation in Nrf2 KO thyroid cells..."). I think this would be clearer if staining for an apical marker such as ezrin or MUC1 are shown and I would also like to see some kind of quantification of follicular organization (e.g. number per area, size) between the WT and KO conditions. I think

that would strengthen the notion of abolished follicular organization in Nrf2 KO cells.

Fig 5A. Also in this image it is difficult to appreciate any genuine "follicular organization" of the WT cells. Again, staining for an apical marker would be desirable. It rather seems like microlumina between two or three cells. Also a close up might be illustrative.

In my opinion, these suggestions seem realistic in terms of time and resources, as this group has established, have access to and are proficient in using both the zebrafish lines as well as the mESC differentiation protocol towards follicular thyroid units.

It is difficult to tell if and how mechanistic insights into the role of Nrf2 in folliculogenesis in the mESC system might be obtained. That might require opening up new lines of experimentation (that I certainly do not require), but I leave it to the authors to judge if some realistic and feasible additional experiments would possibly contribute to more mechanistically oriented understanding. It would be nice to see the authors leverage even more from their beautiful mESC model of thyroid folliculogenesis and I believe that this model might indeed provide important mechanistic insight into this issue.

It is known from previous literature that Tg production is diminished in nrf2 KO mice and that AREs are present in murine Tg enhancer regions (Ziros et al 2018). In the current manuscript the authors do not identify such elements in the vicinity of the zebrafish tg gene. To me, this comparison of AREs in enhancer elements is an important observation that might explain some of the differences to results in Ziros et al and warrants to be included in the results section with a figure, and not only mentioned in the discussion section by referring to a supplementary figure. It would be desirable if the authors elaborate a bit more and clearly on how they envision nrf2a impacts on Tg production in the zebrafish.

In Fig 2L it seems like T4 expression is completely lacking in KO embryos, whereas Fig 2R suggests that a signal that can be quantified is indeed present. Moreover, in Fig 3J a T4 signal, albeit reduced, is seen. Is Fig 2L really representative?

In Fig 3Q the decrease of T4 signal seems much less pronounced than in Fig. 2R, even though it seems like the comparisons are between the same genotypes. Can the authors comment on this?

Fig 6B. What do the two columns in WT and KO represent? Has the experiment been conducted on only two biological replicates?

****Minor comments****

Fig 2S - Do the bars of this graph show the ratio of expression in KO vs WT? What is the black bar furthest to the left (labelled "WT") that seems to be some kind of normalizer? Which transcript does it represent? The same question goes also for 2T-V and 4B and C.

Fig 2U - In the results section it is stated that "Upon PTU treatment, tsh- β and slc5a5a expression were increased in both WT and nrf2a Δ 5 186 homozygous mutants...". In the figure it seems like there is no significant change of slc5a5 in PTU treated homozygous mutants.

Line 52. "...the thyroid enables the production of growth hormones...". This is not clear to me. To the authors mean GH or do they more loosely refer to T3/T4 as "growth hormones"?

Line 60. "... If left untreated, C.H. will cause severe mental and growth retardation in patient among other physiological consequences...". I would consider these consequences as "pathological" rather than "physiological".

Line 151 A.U - if the units are arbitrary, why use such a cumbersome order of magnitude where the numbers are in the order of 10e5 and 10e6?

Line 246. "Based on these results, we hypothesized that adult Nrf2 KO mice might develop body-wide resistance to the effects of Nrf2 deficiency (sic!) which in turn, might reduce the visible effects on thyroid development and physiology". The concept of a putative "body-wide resistance" is a bit nebulous to me. It would be great if the authors

could be a bit more precise, or at least speculate on, the putative mechanisms of such a "body-wide resistance".

Line 297. It is very difficult to appreciate from these images that "the percentage of Nkx2-1 cells was higher compared with the control cell line". In WT it seems that all nuclei are positive for Nkx2-1 but rather that the expression level is lower than in the KO cell line. I would like the authors to elaborate on this. Is really the percentage of "Nkx2-1 cells" (I think the authors mean Nkx2-1 positive cells) lower in WT than in the KO or it is rather a matter of staining intensities?

Line 309. Is it really a "lower portion of cells" that are "able to promote iodide organification"? Do the authors consider some KO cells to be organification competent whereas other cells not? Is it not rather a globally diminished ability to organify iodide?

Line 316. "...KO derived thyroid follicles". This seems contradictory to the previous notion that KO cells do not form follicles. I suggest that "follicles" is replaced by "thyrocytes" if follicular structures are indeed completely lacking. However, the phrasing "KO derived thyroid follicles" suggest that such are indeed present and might be possible to quantify as suggested above.

Line 327 "...among NRF2 WT cells, we detected upregulation...". What do the authors mean by "upregulation" in the context of WT cells? As compared to what? If "upregulated" means as compared to KO cell that does not seem completely appropriate. Even though this might seem like semantics, it is not intuitive to me to describe something as "upregulated" in WT cells, that would rather constitute a baseline condition. Would it rather not be considered as a "downregulation" in KO cells?

Line 337. "... important downregulation" seems a little unorthodox to write in a results section. The downregulation might be significant or not. If it is important or not is a different matter (of subjective biological interpretation, i.e. how biological meaning is appreciated) and more suited to be put into context in the discussion section.

There are scattered typos and grammatical errors that make reading less pleasant and need to be corrected, preferably by a native English speaker.

2. Significance:

This is to the best of my knowledge the first study of a putative role for Nrf2 in thyroid development. However, a role of Nrf2 in thyroid physiology and pathology has previously been rather firmly established. Even though the manuscript is a very nice piece of work, it is perhaps difficult to claim that it in its present form signifies a major conceptual advance of the field, as it provides only limited mechanistic insight, especially with respect to possible "non-canonical" functions of nrf2 (mechanisms of impact on Tg production and folliculogenesis). If such insights could be obtained it would clearly increase the significance of this contribution.

The main strength of the study is that it investigates Nrf2 function in the context of embryonic development. Even though the manuscript makes a point of evolutionary conserved mechanisms, I do not really see that that the discrepancies to results obtained from in vivo studies in higher vertebrates are sufficiently discussed and elaborated on.

Line 240. It feels a bit presumptuous to comment on the contents of an excellent paper that one of the authors of the current manuscript is the senior author of (Ziros et al 2018) and of course knows much better than I do. However, the present manuscript states (about Ziros et al 2018) that "In this study, the authors described how loss of Nrf2 function causes a thyroid phenotype only under stress conditions such as iodine overload. However, in the same study, the authors described that the loss of Nrf2 function causes a reduction of the thyroglobulin expression in PCCL3 rat thyroid follicular cell culture". In my recollection of Ziros et al there is a thyroid phenotype in KO mice not only under stress conditions, but also under normal conditions with reduced TG and increased TG-I? I would like the authors to comment on this.

The authors seem to emphasize the aspect of evolutionary conservation. However, even though I consider the possible effect on folliculogenesis in the mESC model as a very interesting finding, it is difficult to understand if it is a phenomenon that is specific to the mESC model system or of more general importance. As the authors demonstrate, follicles indeed seem to develop in nrf2a deficient zebrafish. In Ziros et al 2018 no images of thyroid morphology are provided, but as KO mice are euthyroid it seems likely that follicular organization is not grossly perturbed. The authors need to elaborate on this. Even if the effect might be more or less specific to the mESC system, that does not

necessarily make it less relevant. It might provide fundamental insights into the process of folliculogenesis, but for greater significance more mechanistic insight would be desirable.

****Referees cross-commenting****

I largely agree with the comments of referee #1, particularly the comment that the title (and some of the discussion) of the paper needs to be changed as pointed out by referee #1 ("...there is no evidence that there is a problem with thyroid development in ZF. The thyroid appears to be enlarged at the end of development, most likely as a consequence of increased TSH stimulation, but there is no developmental defect!").

3. How much time do you estimate the authors will need to complete the suggested revisions:

Between 1 and 3 months

No

Response to Reviewers

Dear Editor,

Thank you for allowing us to submit a revised draft of the manuscript "*Nrf2 promotes thyroid development and hormone synthesis*" to Review Commons. We appreciate and are grateful for the time and effort you and the reviewers dedicated to providing feedback on our manuscript. The insightful comments were very valuable in improving the quality of our paper.

We apologize for the delay in submitting this revised version. Addressing the reviewers' insightful comments required substantial additional experimental work, including new *in vivo* analyses, chromatin accessibility profiling, and functional pathway interrogation. During the revision process, the PhD student who led this project completed her training and left the laboratory, requiring a reorganization of responsibilities and experimental efforts within the team to ensure rigorous completion of the requested studies. We appreciate your patience during this period and believe that the additional data have significantly strengthened the manuscript. In addition, a list of new data and experiments performed is shown below. Please see, in italics, the point-by-point response to the reviewers' comments and concerns. All figures, lines, and page numbers refer to the revised manuscript file.

List of experiments performed following the reviewers' comments:

- Tg staining in zebrafish wt and *nrf2aΔ5* embryos – 6dpf (Supplementary Figure 4).
- Immunostaining for Phalloidin in mESC-derived organoids.
- ATAC sequencing for mESC-derived organoids (WT and Nrf2 KO; day 22).
- AP-1 pathway inhibition and gene/protein expression assessments

Reviewer #1

(Evidence, reproducibility and clarity (Required)):

This paper deals with the role of the transcription factor Nrf2 in the thyroid gland of zebrafish and in a thyroid organoid model. The subject is of relevance since Nrf2 is known to control the cellular response to oxidative stress and the thyroid is an organ where protection of oxidative stress is of major relevance, given the production of reactive oxygen species during thyroid hormone biosynthesis.

The main result is that in Zebrafish (ZF) thyroid Nrf2 appears to be important for thyroid hormone formation since late stages ZF embryos deprived of NRF2 the levels of the thyroid hormone T4 and of its precursor, iodinated thyroglobulin(Tg), are very much decreased. However, there is no significant decrease of Thyroglobulin mRNA, albeit an impairment in the up-regulation of Tg by TSH could be observed. No effect is seen on the structure of the thyroid follicles and no developmental defect is observed, in contrast with the title of the paper.

Conversely, in the mouse thyroid organoid model the absence of Nrf2 results in an impressive decrease of Tg mRNA and in impaired formation of thyroid follicles.

The study is in most part elegant and technically impeccable. The data are well presented and organised as far as figure is concerned. However, much remains to be done on the interpretation and presentation of results. In addition, the text that has been put together sloppily, with many typing and punctuation mistakes and difficult to interpret sentences. A revision of typing and syntax is absolutely needed.

R. Dear reviewer, we appreciate your positive and constructive comments (addressed below) on our manuscript and we apologize for the text clarity and typos. The grammar and text structure were improved following the comments while additional experiments were performed aiming to answer the open questions.

Main concerns:

- 1) The title of the paper needs to be changed. There is no evidence that there is a problem with thyroid development in ZF. The thyroid appears to be enlarged at the end of development, most likely as a consequence of increased TSH stimulation, but there is no developmental defect!

R. Dear reviewer, thank you for pointing it out. Indeed, the initial title did not represent the phenotype observed in both the Zebrafish and murine thyroid organoid models. So, in this revised version the title has been updated to reflect the effect of Nrf2/nrf2a on maturation rather than development. The new title is "The role of Nrf2/nrf2a in thyroid maturation and hormone synthesis in mammalian and non-mammalian models".

- 2) There is an evident contradiction, in ZF, between the marginal, if any, decrease of Tg mRNA and the impressive decrease in T4 and iodinated Tg. This, in my opinion, very interesting discrepancy, is never discussed. Perhaps the authors should look at the level of Tg protein. Is it possible that there is an increased degradation or some negative translational control in absence of Nrf2? Alternatively, is it possible that there is a defect, yet unidentified, in the organification process? Certainly, to conclude, as the authors do in page 11, lines 236-237, that the defect in hormonogenesis depends on thyroglobulin production is, with the data presented, an unproven statement.

R. Dear reviewer, thank you for raising this question and suggesting experimental ways to tackle it. Following your comment, we performed Tg staining in zebrafish embryos. We could observe similar levels of Tg protein in nrf2a ko vs. nrf2a heterozygous and WT. Data have been added as Supplementary Figure 4 and the text has been updated (L. 175-179). This data ruled out our previous hypothesis that TH-impairment would happen in response to lower Tg levels. Since the phenotype observed in nrf2a KO is similar to the one previously demonstrated by our team with the duox KO (Giusti et al., 2020), we hypothesize that duox could be implicated in dyshormonogenesis. Since duox enzymes are known for tightly controlling H2O2 production, an essential factor for T.H. synthesis (Carvalho and Dupuy, 2017), duox dysregulation could further induce oxidative stress, and lead to hypothyroidism (L. 357-365). To assess the duox expression in zebrafish we performed qPCR in pool of embryos and we did not observe any clear change in duox levels in nrf2a KO compared to WT (results included in Fig. 2S). Since duox is not exclusively expressed in the thyroid, we also performed in situ hybridization, however, we never managed to have convincing results using this technique. Finally, despite the observation that tpo and duox genes are expressed in nrf2a KO embryos, we cannot rule out that the activity of those enzymes is preserved and that the T.H. machinery is functional (L. 192-195). Due to the limitations of performing functional assays in zebrafish, the mechanism behind the dyshormonogenesis phenotype is an open question.

- 3) The data on transcriptional effect of NRf2 in the mouse ES cell system do not really add much. Their major effect is to contribute to a lengthy discussion that would really benefit of a substantial reduction.

R. Dear reviewer, for this revised version of the manuscript we included new ATACseq data and combined it with the previously shown transcriptomic to explore the molecular mechanisms by which Nrf2 loss drives such maturation phenotype in mESC-derived thyroid organoids (L. 298-340; figure 6A-H). Using such approach we demonstrate that Nrf2 causes significant changes in chromatin accessibility which is strongly correlated with changes in gene expression profile. We also could demonstrate that Tg expression is indeed impaired by a reduction in chromatin accessibility under the lack of Nrf2, while we identified key pathways/TFs regulated by Nrf2 that could play a role in driving the phenotype. More specifically we identified increased mRNA expression and chromatin accessibility of genes associated with AP-1 pathway activation, such as Fos, Jun-b, and Stats (L. 322-334; Figure 6 and Supplementary Figure 6).

Interestingly, studies have shown that Nrf2 and AP-1-proteins significantly overlap regulating each other at several levels, including transcriptionally. Also, despite Nrf2 being known for binding to ARE and AP-1 to TRE site, they often overlap with AP-1 being embedded into ARE. These tight relations suggest shared feed-

forward and feed-back circuits between NRF2 and AP-1 factors contributing to their functioning. To further investigate if AP-1 overexpression in Nrf2 KO-derived thyroid organoids has a compensatory effect or if it contributes to the phenotype observed, we performed AP-1 inhibition during the thyroid differentiation protocol. Nrf2 KO cells differentiated with the SR11302 inhibitor (from day 7 to 22) partially reduced the Tg mRNA at higher doses (10 μ M) while Tg protein and Tg-I production are not clearly distinct from the control (L. 334-340; Supplementary Figure 7A-B). This suggests that AP-1 upregulation upon loss of Nrf2 might work as a compensatory mechanism, however, due to the lack of Tg expression, which is under a direct effect of Nrf2, the functionality is not recovered.

- 4) More time should be devoted to explain the substantial differences between the three systems studied (two in this paper, one in a previous published by partly same authors), keeping in mind that studies in mice could be largely influenced by the genetic background.

R. Following the suggestion of the reviewer, we added a table (Table 7) summarizing the differences and similarities between the two models used in this study and the in vivo model from Ziros, et al. 2018. We also better discussed the 3 models in the discussion and added subtitles to make it clearer.

Reviewer #1 (Significance (Required)):

This paper is largely confirmatory of previous results obtained in a mouse Nrf2 KO model, whose main authors are also part of this study (Ziros et al., 2018). A clarification of the molecular defect in hormone production in ZF could add the novelty that this study might need.

R. Dear reviewer, indeed our study reinforces the effect of Nrf2 in regulating of Tg expression in mice (not conserved in zebrafish). In addition, we demonstrate the transcriptional and chromatin accessibility changes promoted by the loss of Nrf2 in mouse thyroid cells. Also, the fact that Nrf2 KO ES cells do not efficiently form follicles in vitro is a very interesting and unexpected observation that reinforces the hypothesis that Tg secretion plays a role during early folliculogenesis. In zebrafish, despite the normal expression of the main thyroid markers, the defect in function could be explained by disruptions in duox and tpo activity which would impair Tg iodination. Unfortunately, the limitation of zebrafish for functional studies keeps this question open.

Reviewer #2 (Evidence, reproducibility and clarity (Required)):

Summary

In this manuscript Gillotay et al investigate and further establish a role for Nrf2 in thyroid physiology. Importantly, a role of Nrf2 in thyroid development is investigated in the Tg(tg:nlsEGFP) zebrafish model system developed by this group. This permits detailed and also live tracking of zebrafish thyroid development. By a Crispr/Cas9 approach the authors establish a functional role for nrf2a in zebrafish thyroid development. Nrf2a loss-of-function in zebrafish leads to a hypothyroid phenotype that seems to recapitulate aspects of dyshormonogenesis with a slightly enlarged gland, increased tshb, strongly reduced T4 and iodinated TG (TG-I). Moreover, Tg synthesis is perturbed, only slightly during control conditions but pronouncedly during PTU challenge. By restoring nrf2a specifically in the zebrafish thyroid, the authors prove the phenotype to be cell autonomous to the thyroid.

The authors then turn to the model system of functional thyroid follicular maturation from mESC that they have previously described. In this system Nrf2a KO leads to loss of TG expression and what the authors describe as an inability to form follicular units. Even though Nis is expressed, iodine organification is impaired, likely due to defective Tg production.

Evidence, reproducibility and clarity

Major comments

This is a nicely written manuscript with mostly convincing results. The authors demonstrate laudable scientific rigor by verifying that the genetic modifications indeed have the expected effect (that nrf2 in the model systems that they create is indeed non-functional) and by using a rescue strain for the zebrafish model. In

my opinion the data and the methods are presented in such a way that they can be reproduced. Generally, the conclusions are well supported by the results presented.

The authors make a rather strong point of Nrf2 thyroid function as an evolutionary conserved mechanism and this might need some further underpinning.

R. Dear reviewer, we appreciate your positive comments on our manuscript. Concerning the evolutionary conserved mechanisms driven by Nrf2, we better discuss this aspect in this new version. Even if the lack of nrf2/nrf2a drives the same phenotypical outcome in both models, we discuss the possible distinct mechanisms by which it drives such phenotype. While in mESC-derived thyroid organoids, we confirm the repressive effect of Nrf2 loss on Tg expression and consequently on folliculogenesis and TH production, in zebrafish our new data suggest another mechanism. As suggested by reviewer 1, Tg staining was performed under "physiological" conditions and we did not observe any change in Tg protein in nrf2 KO compared to WT embryos (Supplementary Figure 4). This ruled out the hypothesis of a conserved mechanism involving tg expression. However, as discussed in the new version, this phenotype could be related to dysfunctions in the activity of enzymes associated with tg iodinations, like tpo and duox. However, due to the challenges in performing functional and activity studies in zebrafish, this hypothesis could not be validated.

Line 368. The authors state that their findings "...reinforce a recently published report on the role of Nrf2 in adult mouse thyroid physiology (Ziros et al., 2018)" and continue: "Although we did not analysed (sic!) the thyroid gland of adult nrf2a KO zebrafish"...".

Why were not adult zebrafish investigated? As some key results of Ziros et al on adult Nrf2 KO mice differ from those of the manuscript (e.g. TG iodination under non stressed conditions seems to be increased in mice but decreased in zebrafish) it would be highly desirable to know if the zebrafish phenotype is more similar to that of mice in later adulthood.

R. Dear reviewer, indeed assessing TH status in adult zebrafish would tell us if there is a later compensatory effect occurring, what might be the case in mice. However, despite our will to test it in adults, we faced several challenges: a. A big proportion of the nrf2a homozygous embryos die around 10dpf, which could be a consequence of the absence of THs. b. The few adults that managed to survive have shown much lower ability to produce eggs (20 eggs instead of hundreds), a reduction in fertility. Considering that the complete lack of TH would be detrimental to survival, we could expect that the minority of nrf2a KO embryos able to reach adulthood are the selected ones producing a sufficient amount of T4 to keep them alive. c. Thyroid tissue analysis in adult zebrafish is technically similar to mice, where the tissue dissection and processing is quite challenging since the thyroid is very small and difficult to dissect. Taking into account these limiting aspects, unfortunately, we were not able to provide the TH production assessment for this manuscript.

As I will elaborate on later on, to me a key finding seems to be that nrf2a might have unexpected "non-canonical functions" that do not immediately seem to pertain to its presumed major function as a key regulator of defense to oxidative stress. One of these is to be an impact on Tg production. The other one seems to be a possible role in folliculogenesis, even though this might in some way be related to the impact on Tg production. I do understand that the authors want to leverage from the elegant in vitro model of mESC-derived follicles that they have developed. However, the rationale of using the model system in this context is not entirely clear to me. In Ziros et al the effect of Nrf2 deficiency was studied in a murine global knockout model, in a thyroid specific knockout (even though it might be argued that by using a Pax8 Cre-driver Nrf2 was likely knocked out also in the kidneys and in some regions of the CNS) as well as in cultured rat thyroid cells. Is it the fact that the Pax8 Cre-driver is not entirely thyroid specific that prompted the authors to turn to the mESC-model, the possibility of studying thyroid cell autonomous mechanisms of folliculogenesis (without the possible impact of other tissues and the HPT-axis) by turning to this reductionistic model or other considerations? Even though the results from the mESC model certainly are of interest, the rationale needs to be better explained and the real potential of this model is perhaps not fully exploited. Specifically, the effect on folliculogenesis in the mESC system needs to be more carefully presented:

R. Dear reviewer, indeed, the rationale behind the choice of using mESC-derived thyroid organoids to study Nrf2 loss was not clearly presented in the first version of our manuscript. The text has been modified in the updated version to better explain our choice of using the mESC-derived thyroid model. In fact, the first goal of using this system was to be able to compare the effect of Nrf2 KO during thyroid development in a mammalian system. Compared to in vivo mouse models, assessing/tracking changes (mRNA expression

and live imaging) during early development is challenging due to the need for many animals to study each stage of development. In addition, we aimed to use the derived cells to perform omics and understand the mechanisms behind the phenotype. These experiments would be challenging to perform in zebrafish due to the small number of thyroid cells at 6dpf and the number of cells necessary for RNA and ATAC sequencing. Such assessments were also not performed in the Ziros, 2018 previous study.

To us, in Ziros, 2018 paper, the fact that Pax8 is expressed in the kidney was not a main drawback. In our system, the folliculogenesis phenotype was also not expected, and it is highlighted as an unexpected and novel finding during development, rather than as the rationale for this work. Our new Fig. 5B shows that even if less frequently, smaller follicles are formed. Taking this into account, we cannot exclude that this phenotype could be transitory during early stages of development and that upon Tg accumulation overtime, follicles could be formed and Tg iodinated since the iodine machinery is highly preserved. This hypothesis would fit with the findings from Ziros (2018) that show thyroid follicular organization and function in adult mice lacking Nrf2 expression. Another possible explanation is that during thyroid in vitro differentiation a higher level of stress is expected compared to an in vivo system and in the case of Nrf2 lack it would exacerbate the effect, as previously demonstrated in vivo by the overload of iodine in adult mice (Ziros, 2018).

Line 294. Even though they might be discernible, it is difficult to really appreciate the occurrence of follicular lumina in Fig 4E WT structures. The authors consider this as a main finding (see discussion: "The most striking difference we observed between the two models was the absence of follicular organisation in Nrf2 KO thyroid cells..."). I think this would be clearer if staining for an apical marker such as ezrin or MUC1 are shown and I would also like to see some kind of quantification of follicular organization (e.g. number per area, size) between the WT and KO conditions. I think that would strengthen the notion of abolished follicular organization in Nrf2 KO cells.

R. Dear reviewer, following your comment we performed phalloidin staining to get a better view of the follicular organization in Nrf2 WT and KO-derived thyroid organoids (Fig. 5B). Also, using this staining we could observe that visually fewer and smaller follicle-like structures are formed upon Nrf2 absence. This could indicate that folliculogenesis is not completely abrogated but not fully and properly occurring in this case, mostly likely due to the low accumulation of Tg in the lumen. This reinforces the hypothesis that in mice, overtime accumulation of Tg, even if less produced, could compensate for the folliculogenesis phenotype and finally produce functional follicles (as in Ziros's paper).

Fig 5A. Also in this image it is difficult to appreciate any genuine "follicular organization" of the WT cells. Again, staining for an apical marker would be desirable. It rather seems like microlumina between two or three cells. Also a close up might be illustrative.

R. Dear reviewer, to show more clearly the follicular organization, phalloidin staining has been added to Fig. 5 (B).

In my opinion, these suggestions seem realistic in terms of time and resources, as this group has established, have access to and are proficient in using both the zebrafish lines as well as the mESC differentiation protocol towards follicular thyroid units.

It is difficult to tell if and how mechanistic insights into the role of Nrf2 in folliculogenesis in the mESC system might be obtained. That might require opening up new lines of experimentation (that I certainly do not require), but I leave it to the authors to judge if some realistic and feasible additional experiments would possibly contribute to more mechanistically oriented understanding. It would be nice to see the authors leverage even more from their beautiful mESC model of thyroid folliculogenesis and I believe that this model might indeed provide important mechanistic insight into this issue.

R. Dear reviewer, we fully agree that the unexpected role of Nrf2 in folliculogenesis is something to be better explored in our organoid system. Even though, we are not sure if there is a direct effect of this TF in regulating this process or if it is just a consequence of Tg downregulation. As a future perspective, we aim to keep the Nrf2 KO organoids in culture for longer periods (to be established) and assess if the overtime accumulation of Tg is enough to form bigger and functional follicles.

It is known from previous literature that Tg production is diminished in nrf2 KO mice and that AREs are present in murine Tg enhancer regions (Ziros et al 2018). In the current manuscript the authors do not identify such elements in the vicinity of the zebrafish tg gene. To me, this comparison of AREs in enhancer elements

is an important observation that might explain some of the differences to results in Ziros et al and warrants to be included in the results section with a figure, and not only mentioned in the discussion section by referring to a supplementary figure.

R. Dear reviewer, this aspect was further discussed in the text and the comparison figure was added as part of a main figure (Fig. 6C). In addition, in this updated version we also included new ATAC sequencing data (L. 298-340; Fig. 6) that reinforces the results previously shown by Ziros (2018) in which Nrf2 regulates Tg by binding to the ARE sites in its promoter.

It would be desirable if the authors elaborate a bit more and clearly on how they envision nrf2a impacts on Tg production in the zebrafish.

R. Dear reviewer, as suggested by reviewer 1, we performed immunostaining to evaluate the levels of Tg protein and we did not observe any difference in protein among nrf2a KO and WT embryos (Supplementary Fig. 4). This together with the lack of ARE sites at zebrafish Tg promoted ruled out an effect of nrf2 on its expression in zebrafish under physiological conditions.

In Fig 2L it seems like T4 expression is completely lacking in KO embryos, whereas Fig 2R suggests that a signal that can be quantified is indeed present. Moreover, in Fig 3J a T4 signal, albeit reduced, is seen. Is Fig 2L really representative?

R. Dear reviewer, the displayed pictures are representative of the main phenotype obtained (majority of the embryos). Fluorescent quantification was done using Leica software (described below). The software quantifies the fluorescence based on grayscale images which means that fluorescence slightly higher than the background, thus barely visible, will be quantified which can lead to a value different than zero despite no "visible" staining. This reflects the apparent discordance between the Figures 2L and 2R. Concerning Fig 3J, this is an example that a few nrf2a KO embryos can produce T4 though in lower levels than WT. This range of phenotypes is now highlighted within the text and is reflected in the Fig. 2R and 3Q quantification data.

In Fig 3Q the decrease of T4 signal seems much less pronounced than in Fig. 2R, even though it seems like the comparisons are between the same genotypes. Can the authors comment on this?

R. Dear reviewer, Fig. 2R represents embryos from F2 while Fig. 3Q are from F3 and both were performed independently. Despite the possible differences between the generations, other technical factors could be involved such as: sample fixation, staining duration of antibody, post-processing of the samples, etc. Although we tried to perform both experiments as similarly as possible, we can not rule out small differences between both experiments.

Fig 6B. What do the two columns in WT and KO represent? Has the experiment been conducted on only two biological replicates?

R. Dear reviewer, indeed, the experiments, bulk RNA and ATAC sequencing, we performed using two biological replicates. For each replicated, we pooled together at least 4 organoid wells that were not previously selected, aiming to represent a possible variability in differentiation.

Minor comments

Fig 2S - Do the bars of this graph show the ratio of expression in KO vs WT? What is the black bar furthest to the left (labelled "WT") that seems to be some kind of normalizer? Which transcript does it represent? The same question goes also for 2T-V and 4B and C.

R. Dear reviewer, we apologize for the lack of clarity. All data displays the fold change compared to their respective control (WT). The black bar shows the control (WT=1) for each gene on the figure panel. A dashed line has been added to better visualize the differences in gene expression levels respective to the control. Figure legends have been updated for clarity.

Fig 2U - In the results section it it stated that "Upon PTU treatment, tsh- β and slc5a5a expression were increased in both WT and nrf2a Δ 5 186 homozygous mutants...". In the figure it seems like there is no significant change of slc5a5 in PTU treated homozygous mutants.

R. Dear reviewer, we apologize for the mistake, the sentence was corrected accordingly.

Line 52. "...the thyroid enables the production of growth hormones...". This is not clear to me. To the authors mean GH or do they more loosely refer to T3/T4 as "growth hormones"?

R. Dear reviewer, we agree with the comment and modified the sentence accordingly to precise that the action of the thyroid on growth hormone production is mediated by the thyroid hormones.

Line 60. "... If left untreated, C.H. will cause severe mental and growth retardation in patient among other physiological consequences...". I would consider these consequences as "pathological" rather than "physiological".

R. Dear reviewer, indeed, using pathological is more appropriate, the sentence has been updated.

Line 151 A.U - if the units are arbitrary, why use such a cumbersome order of magnitude where the numbers are in the order of 10^5 and 10^6 ?

R. Dear reviewer, we are performing the fluorescence quantification using the quantification module of the "Leica, LAS X" software. Briefly, we delimitate region of interest for which the software will give us the value of fluorescence for each pixel with this region of interest. Although we are correcting the value of each of these pixels in the region of interest by the average fluorescent value of the pixels in the background area, the amount of pixels in each region is bringing the value to this extent. We decided to keep the raw fluorescence values to better express the differences in magnitude among the groups.

Line 246. "Based on these results, we hypothesized that adult Nrf2 KO mice might develop body-wide resistance to the effects of Nrf2 deficiency (sic!) which in turn, might reduce the visible effects on thyroid development and physiology". The concept of a putative "body-wide resistance" is a bit nebulous to me. It would be great if the authors could be a bit more precise, or at least speculate on, the putative mechanisms of such a "body-wide resistance".

R. Dear reviewer, indeed this is a speculation and we have removed this statement to improve the clarity of the manuscript. In addition, we included in the updated text a hypothesis that the overtime accumulation of Tg from early development (as seen in our in vitro system) to adulthood could lead to the proper formation of thyroid follicles and consequently T.H. synthesis. Even if not included in this manuscript, we plan to improve our in vitro model for long-term culture to assess this hypothesis.

Line 297. It is very difficult to appreciate from these images that "the percentage of Nkx2-1 cells was higher compared with the control cell line". In WT it seems that all nuclei are positive for Nkx2-1 but rather that the expression level is lower than in the KO cell line. I would like the authors to elaborate on this. Is really the percentage of "Nkx2-1 cells" (I think the authors mean Nkx2-1 positive cells) lower in WT than in the KO or it is rather a matter of staining intensities?

R. Dear reviewer, we apologize for the lack of clarity. Rather than referring to the IF images at Fig. 4E we refer to a higher percentage of Nkx2.1+ cells in Nrf2 KO organoids when using Flow cytometry quantification (Fig. 4D). The flow cytometer graphs show the gating for Nkx2.1-stained cells and highlight that upon the absence of Nrf2 65.3% of the cells are Nkx2.1+ compared to 27.1% in the WT controls. We updated the text to avoid misunderstandings and immunostainings are mainly used to show visually the cell organization and protein expression rather than with quantitative purposes.

Line 309. Is it really a "lower portion of cells" that are "able to promote iodide organification"? Do the authors consider some KO cells to be organification competent whereas other cells not? Is it not rather a globally diminished ability to organify iodine?

R. Dear reviewer, we consider that the lower ability of Nrf2 KO cells to produce Tg and consequently self-organize into follicles is the primary cause of the global reduction in iodine organification. Even though, iodine uptake is not impaired, a lower amount of cells can produce Tg-I, thus displaying the ability to organify iodine. Very likely this is limited by the number of Tg-expressing cells and/or the amount of Tg in each cell derived from Nrf2 KO mESCs.

Line 316. "...KO derived thyroid follicles". This seems contradictory to the previous notion that KO cells do not form follicles. I suggest that "follicles" is replaced by "thyrocytes" if follicular structures are indeed completely lacking. However, the phrasing "KO derived thyroid follicles" suggest that such are indeed present and might be possible to quantify as suggested above.

R. Dear reviewer, thank you for pointing it out. Indeed, the term nrf2 KO-derived thyroid follicles is not appropriate and has been changed in the text. Also, since our Phalloidin staining shows that we have small follicles formed we updated our data description and discussion for the fact that follicles seem to form, however a clear delay in size is observed among Nrf2 KO organoids. This is very likely linked to a lower expression of Tg in those organoids.

Line 327 "...among NRF2 WT cells, we detected upregulation...". What do the authors mean by "upregulation" in the context of WT cells? As compared to what? If "upregulated" means as compared to KO cell that does not seem completely appropriate. Even though this might seem like semantics, it is not intuitive to me to describe something as "upregulated" in WT cells, that would rather constitute a baseline condition. Would it rather not be considered as a "downregulation" in KO cells?

R. Dear reviewer, the statement was indeed not appropriate and we modified the text accordingly.

Line 337. "... important downregulation" seems a little unorthodox to write in a results section. The downregulation might be significant or not. If it is important or not is a different matter (of subjective biological interpretation, i.e. how biological meaning is appreciated) and more suited to be put into context in the discussion section.

R. Dear reviewer, following your comments, transcriptomic results and discussion have been updated for clarity. A more factual description has been kept in the result section while the interpretation was moved down to the discussion.

There are scattered typos and grammatical errors that make reading less pleasant and need to be corrected, preferably by a native English speaker.

R. Dear reviewer, we apologize for the text clarity and typos. The grammar and text structure were improved following the comments.

Reviewer #2 (Significance (Required)):

Significance

This is to the best of my knowledge the first study of a putative role for Nrf2 in thyroid development. However, a role of Nrf2 in thyroid physiology and pathology has previously been rather firmly established.

Even though the manuscript is a very nice piece of work, it is perhaps difficult to claim that it in its present form signifies a major conceptual advance of the field, as it provides only limited mechanistic insight, especially with respect to possible "non-canonical" functions of nrf2 (mechanisms of impact on Tg production and folliculogenesis). If such insights could be obtained it would clearly increase the significance of this contribution.

R. Dear reviewer, for this revised version of the manuscript we included a new set of ATACseq data and combined it with the previously shown transcriptomic to further explore the molecular mechanisms by which Nrf2 loss drives such maturation phenotype in developing mESC-derived thyroid organoids (L. 304-340; figure 6A-H). Using such an approach we demonstrate that Nrf2 causes significant changes in chromatin accessibility which is strongly correlated with changes in gene expression profile. We also could demonstrate that Tg expression is indeed impaired by a reduction in chromatin accessibility under the lack of Nrf2, while we identified key pathways/TFs regulated by Nrf2 that could play a role in driving the phenotype or as compensatory mechanisms. More specifically we identified increased mRNA expression and chromatin accessibility of genes associated with AP-1 pathway activation, such as Fos, Jun-b, and Stats (L. 298-333; Figure 6 and Supplementary Figure 6).

Interestingly, studies have shown that Nrf2 and AP-1-proteins significantly overlap regulating each other at several levels, including transcriptionally. Also, despite Nrf2 being known for binding to ARE and AP-1 to the TRE sites, they often overlap with AP-1 being embedded into ARE. These tight relations suggest shared feed-forward and feed-back circuits between NRF2 and AP-1 factors contributing to their functioning. To further investigate if AP-1 overexpression in Nrf2 KO-derived thyroid organoids has a compensatory effect or if it contributes to the phenotype observed, we performed AP-1 inhibition during the thyroid differentiation protocol. Nrf2 KO cells differentiated with the SR11302 inhibitor (from day 7 to 22) partially reduced the Tg mRNA at higher doses (10 μ M) while Tg protein and Tg-I production are not visually different from the control

(L. 334-340; Supplementary Figure 7A-B). This suggests that AP-1 upregulation upon loss of Nrf2 might work as a compensatory mechanism, however, due to the lack of Tg expression, which is under a direct effect of Nrf2, the functionality is not recovered.

The main strength of the study is that it investigates Nrf2 function in the context of embryonic development.

Even though the manuscript makes a point of evolutionary conserved mechanisms, I do not really see that the discrepancies to results obtained from in vivo studies in higher vertebrates are sufficiently discussed and elaborated on.

R. Dear reviewer, following your and reviewer 1 suggestions we have updated our discussion to better highlight the similarities and differences between both models while comparing our mouse in vitro model to the in vivo model previously published.

Line 240. It feels a bit presumptuous to comment on the contents of an excellent paper that one of the authors of the current manuscript is the senior author of (Ziros et al 2018) and of course knows much better than I do.

However, the present manuscript states (about Ziros et al 2018) that "In this study, the authors described how loss of Nrf2 function causes a thyroid phenotype only under stress conditions such as iodine overload. However, in the same study, the authors described that the loss of Nrf2 function causes a reduction of the thyroglobulin expression in PCCL3 rat thyroid follicular cell culture". In my recollection of Ziros et al there is a thyroid phenotype in KO mice not only under stress conditions, but also under normal conditions with reduced TG and increased TG-I? I would like the authors to comment on this.

R. Dear reviewer, indeed, in Ziros' (2018) paper it is shown that lack of Nrf2 does not cause hypothyroidism in physiological conditions, however, Tg expression is reduced (regulation mechanisms were also shown) but the Tg-I/Tg iodination rate was increased. Only under iodine overload, they could see an inhibitory effect on thyroid hormone production. We were not clear in our statement in the previous version which is now improved in the revised version. In addition, we also better discuss our findings in organoids and raise the hypothesis that long-term Tg accumulation could "restore" folliculogenesis and thyroid hormone synthesis.

The authors seem to emphasize the aspect of evolutionary conservation. However, even though I consider the possible effect on folliculogenesis in the mESC model as a very interesting finding, it is difficult to understand if it is a phenomenon that is specific to the mESC model system or of more general importance. As the authors demonstrate, follicles indeed seem to develop in nrf2a deficient zebrafish. In Ziros et al 2018 no images of thyroid morphology are provided, but as KO mice are euthyroid it seems likely that follicular organization is not grossly perturbed. The authors need to elaborate on this. Even if the effect might be more or less specific to the mESC system, that does not necessarily make it less relevant. It might provide fundamental insights into the process of folliculogenesis, but for greater significance more mechanistic insight would be desirable.

R. Dear reviewer, as mentioned above, we used new tools to better analyze the "folliculogenesis impairment" previously suggested to be occurring in our organoid system. A careful assessment of the morphology of our Nrf2 KO-derived organoids using Phalloidin (Fig. 5B) staining evidenced that in fact folliculogenesis process might be undergoing in our organoids, however, the follicular-like structures are less frequently observed (difficult to quantify due to the 3D aspect of the follicular organization) while the size seems to be smaller than in WT organoids. Here we believe that due to the lower levels of Tg expressed and secreted into the lumen the size is smaller. This proposed hypothesis fits with the previous studies suggesting that Tg has a role in folliculogenesis. In addition, we cannot rule out that in vivo this same phenotype happens during early development and that overtime accumulation of Tg could lead to proper follicular formation and consequently to normal thyroid function. Interestingly, Ziros' paper shows that even in Tg downregulation conditions, T.H. production is not impaired, with a higher ratio of iodinated Tg compared to WT mice, suggesting a compensatory mechanism to overcome the lower levels of Tg. This new aspect is now further discussed in the manuscript.

****Referees cross-commenting****

I largely agree with the comments of referee #1, particularly the comment that the title (and some of the discussion) of the paper needs to be changed as pointed out by referee #1 ("...there is no evidence that there is a problem with thyroid development in ZF. The thyroid appears to be enlarged at the end of development, most likely as a consequence of increased TSH stimulation, but there is no developmental defect!").

R. Dear reviewer, following both reviewer's suggestions, we modified the title of the paper to better reflect the results presented. The new title is "The role of Nrf2/nrf2a in thyroid maturation and hormone synthesis in mammalian and non-mammalian models".

In humans, developmental defects such as congenital hypothyroidism can be divided into two main categories: 1. Dyshormonogenesis, when the tissue is properly developed but thyroid function is impaired and 2. Dysgenesis, when the tissue (organogenesis is impaired) is not properly formed or not at all, resulting in hypothyroidism. The phenotype observed in nrf2a KO zebrafish corresponds to the dishormonogenesis in humans and despite that folliculogenesis seems to be preserved, the bigger size of the thyroid is not considered as a defect in organogenesis but a consequence of higher TSH stimulation. Still, it would be classified as a developmental defect. However, to avoid misinterpretation, we updated the text and highlighted that in zebrafish the lack of nrf2a results in hyperplastic non-functional thyroid tissue.

April 6, 2026

RE: Life Science Alliance Manuscript #LSA-2026-03687

Sabine Costagliola

Dear Dr. Costagliola,

Thank you for submitting your revised manuscript entitled "The role of Nrf2/nrf2a in thyroid maturation and hormone synthesis in mammalian and non-mammalian models". Your manuscript was evaluated by the original reviewers at Review Commons and their comments are below. As you will see, the reviewers are satisfied overall and recommend publication of this work. Please attend to the remaining minor points raised by Reviewer 2, on annotating hair follicles in the images provided and on some improvements to the text. We would be happy to publish your paper in Life Science Alliance pending these changes and final revisions necessary to meet our formatting guidelines.

MANUSCRIPT ORGANIZATION AND FORMATTING:

To avoid unnecessary delays in the acceptance and publication of your paper, please read the following information carefully. Full guidelines are available on our Instructions for Authors page, <https://www.life-science-alliance.org/authors>

- Please ensure the titles in the manuscript file and in our submission system match. We appreciate that title in the manuscript file was changed to satisfy the concern of Reviewer 1. Unfortunately this title does not succinctly capture the main findings of this work. Please amend the title to state the role for Nrf2 that was found (thyroid hormone production) and refer to vertebrate models instead of "mammalian and non-mammalian models".
- Please consider refocusing the abstract on the main results found while reducing emphasis on the background, for instance by removing the second sentence.
- Please be sure that the authorship listing and order are correct and match between the system and manuscript file.
- Please add a Running Title in our system.
- Please add a Summary Blurb/Alternate Abstract in our system.
- Please add an Author Contributions section to your main manuscript text.
- Please add a Conflict of Interest statement to your main manuscript text.
- Please upload all figure files as individual files, including the supplementary figure files; all figure legends should only appear in the main manuscript file.
- Please add callouts for Figure 2 A-O, Figure 3R, Figure 5E-F, Figure 6H, Figure 7, all panels of Supplemental Figures 2, 3, 4, 6, and 7, and the supplemental Table, to your main manuscript text; -LSA allows supplementary figures, but not EV Figures; please update your callouts for the Supplementary Figures in the manuscript Fig EV1A = Fig S1A)
- LSA does not permit citation of "data not shown," "manuscript in preparation," "manuscript submitted," etc., in any section of the manuscript. Please either remove these claims on pages 5, 11, 16, 17, and 36 or include the cited data.

We welcome submissions of potential cover images for the issue of LSA in which your work would appear. If you have high quality images associated with this work, please feel free to email these, with a caption, to the journal office.

LSA encourages authors to provide a 30-60 second video where the study is briefly explained. We will use these videos on social media to promote the published paper and the presenting author (for examples, see <https://docs.google.com/document/d/1-UWCfbE4pGcDdcgzcmiuJI2XMBJnxKYeqRvLLrLSo8s/edit?usp=sharing>). Corresponding or first-authors are welcome to submit the video. Please submit only one video per manuscript. The video can be emailed to contact@life-science-alliance.org

FINAL FILES:

The following items are required for acceptance.

The license to publish form must be signed before your manuscript can be sent to production. A link to the license to publish form will be available to the corresponding author only. Please take a moment to check your funder requirements.

Thank you for your attention to these final processing requirements. Please revise and format the manuscript and upload materials as soon as you are able.

Thank you for this interesting contribution to the literature. We look forward to publishing your paper in Life Science Alliance.

Sincerely,

Reviewer #1 (Comments to the Authors (Required)):

This paper deals with the role of the transcription factor Nrf2 in the thyroid of two different organisms. The authors report that Nrf2 plays different roles in the thyroid of zebrafish and mice. While in zebrafish the absence of Nrf2 appears to impair thyroid hormone formation, via a decrease in Thyroglobulin iodination, in mice the absence of Nrf2 causes a large decrease in Thyroglobulin mRNA and an impairment in the formation of thyroid follicles. These observations are of relevance since Nrf2 is known to control oxidative stress, but no information was available on which level Nrf2 exerts its action in thyroid, an organ in which protection from oxidative stress is known to be of major relevance.

I have already reviewed this paper in the past and I expressed three main concerns: 1) The title of the paper did not reflect appropriately its content. This has been corrected and the title now is perfect; 2) The impairment in thyroid hormone production in the Zebrafish model was not investigated in depth. It was proposed, not convincingly, that there was a decrease in Thyroglobulin protein. Now solid data demonstrates that it is not the level of thyroglobulin protein that is decreased but its iodination. The mechanism responsible for such a role of Nrf2 in thyroglobulin iodination remains to be clarified, due to objective limitation in assaying the activity of the iodination system in zebrafish. I do think, however, that for this study the definition of the molecular defect is convincingly demonstrated, albeit not in every detail. 3) In the new version much progress has been made toward a mechanistic explanation of the decrease of Thyroglobulin gene expression in the mouse model, involving a decrease in chromatin accessibility. Also the interplay between Nrf2 and Ap-1 are interesting and well documented. Finally, the authors took good care of the manuscript presentation, which was not done in the previous version.

Reviewer #2 (Comments to the Authors (Required)):

In this manuscript Gillotay et al investigate and further establish a role for Nrf2 in thyroid physiology. Importantly, a role of Nrf2 in

thyroid development is investigated in a Tg(tg:nlsEGFP) zebrafish model system developed by this group. This permits detailed and also live tracking of zebrafish thyroid development. By a Crispr/Cas9 approach the authors establish a functional role for nrf2a in zebrafish thyroid development. Nrf2a loss-of-function in zebrafish leads to a hypothyroid phenotype that seems to recapitulate aspects of dyshormonogenesis with a slightly enlarged gland, increased tshb, strongly reduced T4 and iodinated TG (TG-I) whereas Tg expression is largely unaffected. By restoring nrf2a specifically in the zebrafish thyroid, the authors prove the phenotype to be cell autonomous to the thyroid. The authors then turn to the model system of functional thyroid follicular maturation from mESC that they have previously described. In this system Nrf2a KO leads to loss of TG expression and what the authors describe as an inability to form follicular units. Even though Nis is expressed, iodine organification is impaired, likely due to defective Tg production. Finally, the authors investigate how Nrf2 affects the transcriptional and chromatin accessibility landscape and based on this direct their interest towards AP-1.

In my opinion the authors have adequately addressed issues previously raised by me as well as by other reviewers. Relevant data has been added and I think the manuscript has been improved as a whole. I appreciate the summary table (figure 7), that outlines the similarities and differences between zebrafish, mESCs and mice. The manuscript does not provide full mechanistic explanations to these differences, but I think the authors have made a good effort in that direction that satisfies my expectations. I think a strength of this manuscript is that it actually highlights the differences between three major experimental systems when knocking out the same gene. That should not at all be considered a deficiency of the manuscript, but rather serve as a good example of the value of interrogating different systems before reaching definitive conclusions. This makes the manuscript of interest not only to the thyroid community, but to a broader audience. Taken together, this is a fine manuscript and after addressing the issues below I think it can be accepted for publication in Life Science Alliance.

My main concern is that to discuss defective follicle formation in the mESC model, a little more quantitative underpinning is needed. I do not put into question that there really is a follicle formation phenotype, but the authors need to show this more convincingly. The morphology in different images of mESC WT and KO cultures seems to be rather variable and some more quantification seems to be warranted. I do appreciate the new images with phalloidin stains. From the images (Fig 5B) it is however not immediately obvious what the authors consider to be follicles. Some annotation (arrows etc) in these images is very much needed.

The authors write that:

- "A closer look at the follicular organization using Phalloidin immunostaining revealed that Nrf2 KO-derived thyrocytes display a reduced ability to form follicles while they appeared to be smaller than the control-derived ones (Fig. 5C)" ("while" makes this sentence really unclear to me)
- "Immunofluorescence analysis revealed an impairment of our Nrf2 KO cell line to efficiently generate thyroid follicles, even if less frequently also shown to form smaller follicular-organized structures, defined by the presence of a lumen (Fig. 4E and 5B). (also awkward sentence and syntax)
- "Morphologically, mESC Nrf2 KO-derived organoids showed disrupted follicular organization, with visually fewer and smaller follicles..."

In figure 5C, the authors quantify follicle size and show that follicles are smaller in the KO. There is a highly significant difference between WT and KO, but perhaps the actual size difference does not seem to be very impressive. More importantly, the authors need to show (i.e. quantify in an appropriate way) that the number of follicles is indeed reduced (not that they are just smaller), since the authors make the follicle phenotype a major finding of the manuscript. Is it really an initial early step of follicle formation that is impaired, or is it follicle lumen expansion (or both)? This needs to be more clearly shown and discussed.

A few minor issues:

I can't see that phalloidin staining is described in Materials and Methods. Moreover, phalloidin staining is referred to as "Phalloidin immunostaining" (line 297 and possibly also elsewhere). Is that really an immunostaining (with an antibody against phalloidin)? I suspect that it is not an antibody based staining, but that fluorochrome conjugated phalloidin has been used. This needs to be sorted out.

In Fig 2S one bar in "slc5a5" whereas in Fig 2T one is "nis". Why? Is it the same transcript that is quantified? I think slc5a5 is the gene symbol and nis the protein. Would be good if this is made consequent, but I might misunderstand something.

In Fig 2V ratios (KO to WT) are shown. I understand that the tshb ratio is > 1 (KO>WT) and that the tg ratio is < 1 (KO) as the KO expression level is less than the WT expression level. Could it be that it is instead the WT to KO ratio that is erroneously shown? Again, this might be some misunderstanding on my side and if so I am sorry.

RE: Life Science Alliance Manuscript #LSA-2026-03687

Dear Dr. Costagliola,

Thank you for submitting your revised manuscript entitled "The role of Nrf2/nrf2a in thyroid maturation and hormone synthesis in mammalian and non-mammalian models". Your manuscript was evaluated by the original reviewers at Review Commons and their comments are below. As you will see, the reviewers are satisfied overall and recommend publication of this work. Please attend to the remaining minor points raised by Reviewer 2, on annotating hair follicles in the images provided and on some improvements to the text. We would be happy to publish your paper in Life Science Alliance pending these changes and final revisions necessary to meet our formatting guidelines.

R. Dear Editor, we thank you for the positive evaluation of our revised manuscript, as well as for the opportunity to further improve the work for publication in *Life Science Alliance*. We are pleased that the reviewers are overall satisfied and supportive of publication.

We confirm that we have now addressed the remaining minor comments raised by Reviewer #2. In particular, we have updated the figure annotations as requested and made the corresponding clarifications and improvements to the text to ensure clarity and consistency throughout the manuscript.

We have also carefully reviewed all formatting requirements and adjusted the manuscript accordingly to comply with the journal guidelines.

We look forward to the final evaluation and thank you again for handling our submission.

MANUSCRIPT ORGANIZATION AND FORMATTING:

To avoid unnecessary delays in the acceptance and publication of your paper, please read the following information carefully. Full guidelines are available on our Instructions for Authors page, <https://www.life-science-alliance.org/authors>

-Please ensure the titles in the manuscript file and in our submission system match. We appreciate that title in the manuscript file was changed to satisfy the concern of Reviewer 1. Unfortunately this title does not succinctly capture the main findings of this work. Please amend the title to state the role for Nrf2 that was found (thyroid hormone production) and refer to vertebrate models instead of "mammalian and non-mammalian models".

R. We have amended the title accordingly to address the editor's suggestion.

-Please consider refocusing the abstract on the main results found while reducing emphasis on the background, for instance by removing the second sentence.

R. We have modified the abstract following the editor's suggestion.

-Please be sure that the authorship listing and order are correct and match between the system and manuscript file.

-Please add a Running Title in our system.

R. Running title is now included in the revised version.

-Please add a Summary Blurb/Alternate Abstract in our system.

R. Here is the summary blurb: "The thyroid produces hormones in a high oxidative stress environment. Nrf2 is a key antioxidant regulator, and its loss impairs thyroid hormone production in zebrafish and mouse organoids."

-Please add an Author Contributions section to your main manuscript text.

R. The Authors Contributions are now described in the main manuscript text.

-Please add a Conflict of Interest statement to your main manuscript text.

R. The Conflict of Interest statement was included in the main manuscript text.

-Please upload all figure files as individual files, including the supplementary figure files; all figure legends should only appear in the main manuscript file.

-Please add callouts for Figure 2 A-O, Figure 3R, Figure 5E-F, Figure 6H, Figure 7, all panels of Supplemental Figures 2, 3, 4, 6, and 7, and the supplemental Table, to your main manuscript text; -LSA allows supplementary

figures, but not EV Figures; please update your callouts for the Supplementary Figures in the manuscript Fig EV1A = Fig S1A)

R. All the required actions were taken and the information is properly corrected/inserted in the text.

-LSA does not permit citation of "data not shown," "manuscript in preparation," "manuscript submitted," etc., in any section of the manuscript. Please either remove these claims on pages 5, 11, 16, 17, and 36 or include the cited data.

R. All changes were made accordingly.

We welcome submissions of potential cover images for the issue of LSA in which your work would appear. If you have high quality images associated with this work, please feel free to email these, with a caption, to the journal office.

LSA encourages authors to provide a 30-60 second video where the study is briefly explained. We will use these videos on social media to promote the published paper and the presenting author (for examples, see <https://docs.google.com/document/d/1-UWCfbE4pGcDdcgzcmiuJl2XMBJnxKYegRvLLrLSo8s/edit?usp=sharing>). Corresponding or first-authors are welcome to submit the video. Please submit only one video per manuscript. The video can be emailed to contact@life-science-alliance.org

FINAL FILES:

The following items are required for acceptance.

The license to publish form must be signed before your manuscript can be sent to production. A link to the license to publish form will be available to the corresponding author only. Please take a moment to check your funder requirements.

Thank you for your attention to these final processing requirements. Please revise and format the manuscript and upload materials as soon as you are able.

Thank you for this interesting contribution to the literature. We look forward to publishing your paper in Life Science Alliance.

Sincerely,

Reviewer #1 (Comments to the Authors (Required)):

This paper deals with the role of the transcription factor Nrf2 in the thyroid of two different organisms. The authors report that Nrf2 plays different roles in the thyroid of zebrafish and mice. While in zebrafish the absence of Nrf2 appears to impair thyroid hormone formation, via a decrease in Thyroglobulin iodination, in mice the absence of Nrf2 causes a large decrease in Thyroglobulin mRNA and an impairment in the formation of thyroid follicles. These observations are of relevance since Nrf2 is known to control oxidative stress, but no information was available on which level Nrf2 exerts its action in thyroid, an organ in which protection from oxidative stress is known to be of major relevance.

I have already reviewed this paper in the past and I expressed three main concerns: 1) The title of the paper did not reflect appropriately its content. This has been corrected and the title now is perfect; 2) The impairment in thyroid hormone production in the Zebrafish model was not investigated in depth. It was proposed, not convincingly, that there was a decrease in Thyroglobulin protein. Now solid data demonstrates that it is not the level of thyroglobulin protein that is decreased but its iodination. The mechanism responsible for such a role of Nrf2 in thyroglobulin iodination remains to be clarified, due to objective limitation in assaying the activity of the iodination system in zebrafish. I do think, however, that for this study the definition of the molecular defect is convincingly demonstrated, albeit not in every detail. 3) In the new version much progress has been made toward a mechanistic explanation of the decrease of Thyroglobulin gene expression in the mouse model, involving a decrease in chromatin accessibility. Also the interplay between Nrf2 and Ap-1 are interesting and well documented. Finally, the authors took good care of the manuscript presentation, which was not done in the previous version.

R. We would like to sincerely thank Reviewer #1 for the careful and constructive evaluation of our revised manuscript, as well as for the insightful comments provided during both rounds of review. We greatly appreciate the time and effort dedicated to improving the clarity and strength of our study.

We are pleased that the reviewer considers the revised title appropriate and reflective of the content, and that the updated zebrafish data now convincingly demonstrate a defect in thyroglobulin iodination rather than protein abundance. We also appreciate the positive assessment of the mechanistic advances made in the mouse model, particularly the evidence linking Nrf2 deficiency to reduced chromatin accessibility at the Tg locus and the proposed interplay between Nrf2 and AP-1, which we agree adds important mechanistic depth to our findings.

We are equally grateful for the positive comments on the improved presentation and overall quality of the manuscript. We thank the reviewer again for the constructive feedback that has helped us significantly strengthen the study.

Reviewer #2 (Comments to the Authors (Required)):

In this manuscript Gillotay et al investigate and further establish a role for Nrf2 in thyroid physiology. Importantly, a role of Nrf2 in thyroid development is investigated in a Tg(tg:nlsEGFP) zebrafish model system developed by this group. This permits detailed and also live tracking of zebrafish thyroid development. By a Crispr/Cas9 approach the authors establish a functional role for nrf2a in zebrafish thyroid development. Nrf2a loss-of-function in zebrafish leads to a hypothyroid phenotype that seems to recapitulate aspects of dysmorphogenesis with a slightly enlarged gland, increased tshb, strongly reduced T4 and iodinated TG (TG-I) whereas Tg expression is largely unaffected. By restoring nrf2a specifically in the zebrafish thyroid, the authors prove the phenotype to be cell autonomous to the thyroid. The authors then turn to the model system of functional thyroid follicular maturation from mESC that they have previously described. In this system Nrf2a KO leads to loss of TG expression and what the authors describe as an inability to form follicular units. Even though Nis is expressed, iodine organification is impaired, likely due to defective Tg production. Finally, the authors investigate how Nrf2 affects the transcriptional and chromatin accessibility landscape and based on this direct their interest towards AP-1.

In my opinion the authors have adequately addressed issues previously raised by me as well as by other reviewers. Relevant data has been added and I think the manuscript has been improved as a whole. I appreciate the summary table (figure 7), that outlines the similarities and differences between zebrafish, mESCs and mice. The manuscript does not provide full mechanistic explanations to these differences, but I think the authors have made a good effort in that direction that satisfies my expectations. I think a strength of this manuscript is that it actually highlights the differences between three major experimental systems when knocking out the same gene. That should not at all be considered a deficiency of the manuscript, but rather serve as a good example of the value of interrogating different systems before reaching definitive conclusions. This makes the manuscript of interest not only to the thyroid

community, but to a broader audience. Taken together, this is a fine manuscript and after addressing the issues below I think it can be accepted for publication in Life Science Alliance.

R. We would like to sincerely thank Reviewer #2 for the positive and constructive evaluation of our revised manuscript. We greatly appreciate the reviewer's careful assessment of our work and the recognition of the overall improvements made during revision.

Below, we address the remaining comments, and we believe that the clarifications and corresponding changes satisfactorily resolve the remaining unclear points and further strengthen the manuscript for publication.

My main concern is that to discuss defective follicle formation in the mESC model, a little more quantitative underpinning is needed. I do not put into question that there really is a follicle formation phenotype, but the authors need to show this more convincingly. The morphology in different images of mESC WT and KO cultures seems to be rather variable and some more quantification seems to be warranted. I do appreciate the new images with phalloidin stains. From the images (Fig 5B) it is however not immediately obvious what the authors consider to be follicles. Some annotation (arrows etc) in these images is very much needed.

R. Dear reviewer, we thank you for your insightful comments. Regarding the folliculogenesis phenotype observed in Nrf2-KO cells, we have carefully re-examined our data and revised the manuscript accordingly to provide a clearer and more accurate description of our findings. In the updated version, we emphasize the variable phenotype associated with Nrf2-KO cultures, which includes both disorganized structures and follicles that are heterogeneous in shape and size (now indicated by arrows in Fig. 5B). We have also included below an additional figure to better illustrate this variability in follicular organization.

With respect to the quantification of follicle formation efficiency, we revisited our imaging data. However, due to the markedly increased proportion of Nkx2.1⁺ cells in Nrf2-KO cultures (approximately threefold higher than controls) and the intrinsic complexity of 3D structures, accurate quantification of follicle formation across conditions was not feasible. Consequently, we have revised our conclusions to focus on the qualitative differences in follicular organization rather than on follicle formation efficiency.

Finally, we note that this phenotype may be related to the reduced thyroglobulin (Tg) content observed in Nrf2-KO cells, which could impair follicle formation or influence luminal size, as discussed in the revised Discussion section.

Figure 1. Thyroid organoids derived from Nrf2 WT and KO cells show distinct follicular formation phenotype. While in WT-derived organoids most of cells organize into follicles with quite homogeneous organization, Nrf2 KO-derived cells show regions with no clear follicular organization and others where cells clearly organize into follicular structures. Phalloidin rings (yellow) mark the apical membrane of the follicles and highlight the luminal compartment.

The authors write that:

- "A closer look at the follicular organization using Phalloidin immunostaining revealed that Nrf2 KO-derived thyrocytes display a reduced ability to form follicles while they appeared to be smaller than the control-derived ones (Fig. 5C)" ("while" makes this sentence really unclear to me)
- "Immunofluorescence analysis revealed an impairment of our Nrf2 KO cell line to efficiently generate thyroid follicles, even if less frequently also shown to form smaller follicular-organized structures, defined by the presence of a lumen (Fig. 4E and 5B). (also awkward sentence and syntax)"

R. Following the reviewer's comment and our previous response, we have revised the text for clarity. The updated version is now included in the manuscript:

L. 277-288: "Despite the increased proportion of Nkx2.1⁺ cells (Fig. 4D), closer examination of follicular organization using phalloidin immunostaining revealed qualitative differences in follicular architecture in Nrf2 KO cultures. Compared to controls, Nrf2 KO-derived cells exhibit both disorganized and follicular arrangements; however, the follicles that do form are more heterogeneous in size and shape and are often smaller. These observations indicate increased morphological variability in follicle organization under microscopic examination; however, due to technical limitations inherent to 3D cultures, precise quantification is not feasible, precluding definitive conclusions regarding the efficiency or overall integrity of folliculogenesis (Fig. 4E and 5B-C)."

L. 295-297: "This impairment likely reflects both decreased Tg expression and the altered, more variable follicular organization observed under Nrf2 KO conditions (Fig. 4E and 5B-C)."

- "Morphologically, mESC Nrf2 KO-derived organoids showed disrupted follicular organization, with visually fewer and smaller follicles..."

R. To keep consistent with the previous modifications, the sentence was also updated in the text:

L. 409-411: "Morphologically, mESC Nrf2 KO-derived organoids exhibited increased variability in follicular organization, with generally smaller follicles compared to controls, a phenotype that was not observed in zebrafish embryos."

In figure 5C, the authors quantify follicle size and show that follicles are smaller in the KO. There is a highly significant difference between WT and KO, but perhaps the actual size difference does not seem to be very impressive. More importantly, the authors need to show (i.e. quantify in an appropriate way) that the number of follicles is indeed reduced (not that they are just smaller), since the authors make the follicle phenotype a major finding of the manuscript. Is it really an initial early step of follicle formation that is impaired, or is it follicle lumen expansion (or both)? This needs to be more clearly shown and discussed.

R. Regarding follicle size quantification, we analyzed multiple images from Nrf2 WT and KO-derived organoids stained for Nkx2.1 and phalloidin to delineate lumens and measure follicular size. The resulting data show variability in follicle size in both conditions; however, on average, Nrf2 KO-derived follicles are smaller. This effect may be related to reduced Tg levels within the lumen, although we consider the observation robust and have therefore retained it.

With respect to follicle formation efficiency, as described above and now clarified in the main text, the complexity of the 3D system and the limited precision of available measurements precluded reliable quantification. We therefore focused our conclusions on qualitative differences in follicular organization and removed statements implying impaired follicle formation efficiency.

A few minor issues:

I can't see that phalloidin staining is described in Materials and Methods. Moreover, phalloidin staining is referred to as "Phalloidin immunostaining" (line 297 and possibly also elsewhere). Is that really an immunostaining (with an antibody against phalloidin)? I suspect that it is not an antibody based staining, but that fluorochrome conjugated phalloidin has been used. This needs to be sorted out.

R. We apologize for this oversight. We have now added the relevant information on phalloidin staining to the Materials and Methods section. Specifically, we used Alexa Fluor 647-conjugated phalloidin (1:100, cat. A22287, Invitrogen) for F-actin labeling. We have also corrected the terminology throughout the manuscript, replacing "phalloidin immunostaining" with "phalloidin staining" to accurately reflect that no antibody-based detection was used.

In Fig 2S one bar in "slc5a5" whereas in Fig 2T one is "nis". Why? Is it the same transcript that is quantified? I think slc5a5 is the gene symbol and nis the protein. Would be good if this is made consequent, but I might misunderstand something.

R. We apologize for the confusion. You are correct: *Slc5a5* is the gene symbol, while *Nis* refers to the corresponding protein. In the original version, Fig. 2T was incorrectly labeled. We have now corrected this inconsistency and updated the figure in the revised manuscript to ensure consistent nomenclature throughout.

In Fig 2V ratios (KO to WT) are shown. I understand that the tshb ratio is > 1 (KO>WT) and that the tg ratio is < 1 (KO<WT). I do not understand though that the slc5a5 ratio is >1 as the KO expression level is less than the WT expression level. Could it be that it is instead the WT to KO ratio that is erroneously shown? Again, this might be some misunderstanding on my side and if so I am sorry.

R. We thank the reviewer for carefully examining Figure 2V and for raising this point. We have re-checked the calculations and confirm that the graph and ratios are correct as presented. We agree that the apparent inversion arises from differences in baseline expression levels between genotypes, and we have now clarified this explanation in the revised manuscript text and figure legend.

In Figure 2, we performed qPCR analysis of thyroid function–related genes in zebrafish embryos. Analyses were conducted on both wild-type (WT) and *nrf2a* homozygous mutant embryos treated with PTU or vehicle control (DMSO), using three biological replicates and three technical replicates per biological replicate.

In Figure 2S, we compared gene expression between non-treated WT and non-treated *nrf2a* mutant embryos. This analysis revealed that *nrf2a* knockout embryos exhibit higher basal expression levels of *tshb*, *slc5a5*, and *tpo* compared to WT embryos. Notably, *nrf2a* mutant animals show approximately a 4–5-fold higher basal expression of *slc5a5* relative to WT.

In Figures 2T and 2U, we compared the fold-change induction of thyroid-related genes in WT and *nrf2a* mutant animals following PTU treatment relative to their respective non-treated controls. The purpose of these analyses was to demonstrate that PTU treatment can still induce thyroid gene expression in *nrf2a* mutant embryos by inhibiting residual thyroid hormone production, thereby indicating that the thyroid gland in *nrf2a* mutants remains responsive to TSH stimulation.

In Figure 2V, we directly compared gene expression levels between PTU-treated WT and PTU-treated *nrf2a* mutant embryos. The ratio may appear inverted when compared to the fold-change graphs in Figures 2T and 2U because those earlier panels display relative fold induction within each genotype, whereas Figure 2V shows a direct comparison between genotypes under the same treatment condition.

Importantly, the magnitude of fold change depends on the initial baseline expression level. Thus, although the fold-change response to PTU appears larger in WT than in *nrf2a* mutants (4.5 vs 2), the absolute expression level of *slc5a5* remains higher in *nrf2a* mutants under both basal and PTU-treated conditions. Consequently, when directly comparing PTU-treated groups (Figure 2V), *nrf2a* mutants show approximately a twofold higher expression than WT, which is correctly represented as the KOWT ratio in the figure.

To prevent confusion, we have now:

1. Re-checked all calculations and confirmed their correctness
2. Clarified in the revised text and legend that Figures 2T–U represent within-genotype fold induction, whereas Figure 2V represents between-genotype comparison
3. Explicitly stated that differences in baseline expression account for the apparent discrepancy in fold-change magnitude

We hope this clarification resolves the reviewer's concern.

April 24, 2026

RE: Life Science Alliance Manuscript #LSA-2026-03687R

Ms. Sabine Costagliola
Université Libre de Bruxelles
IRIBHM JE Dumont
Belize

Dear Dr. Costagliola,

Thank you for submitting your Research Article entitled "The role of Nrf2 in thyroid maturation and hormone synthesis in vertebrate models". It is a pleasure to let you know that your manuscript is now accepted for publication in Life Science Alliance. Congratulations on this interesting work.

Your article will publish open access upon publication under a CC-BY license.

DISTRIBUTION OF MATERIALS:

Again, congratulations on a very nice paper. I hope you found the review process to be constructive and are pleased with how the manuscript was handled editorially. We look forward to future exciting submissions from your lab.

Sincerely,
